# Analyzing the fine structure of distributions

**Michael C. Thrun**[1,2]\*, **Tino Gehlert**[3], **Alfred Ultsch**[1]

**1** Databionics AG, Dept. of Mathematics and Computer Science, Philipps-University of Marburg, Marburg, Germany, **2** Dept. of Hematology, Oncology and Immunology, Philipps-University Marburg, Germany, **3** Alumni of Faculty of Mathematics, Chemnitz University of Technology, Chemnitz, Germany

\* m.thrun@informatik.uni-marburg.de

**Data Availability Statement:** Data is attached to packages in R: https://CRAN.R-project.org/package=DataVisualizations and in python: https://pypi.org/project/md-plot/

**Funding:** The authors received no specific funding for this work.

## Abstract

One aim of data mining is the identification of interesting structures in data. For better analytical results, the basic properties of an empirical distribution, such as skewness and eventual clipping, i.e. hard limits in value ranges, need to be assessed. Of particular interest is the question of whether the data originate from one process or contain subsets related to different states of the data producing process. Data visualization tools should deliver a clear picture of the univariate probability density distribution (PDF) for each feature. Visualization tools for PDFs typically use kernel density estimates and include both the classical histogram, as well as the modern tools like ridgeline plots, bean plots and violin plots. If density estimation parameters remain in a default setting, conventional methods pose several problems when visualizing the PDF of uniform, multimodal, skewed distributions and distributions with clipped data, For that reason, a new visualization tool called the mirrored density plot (MD plot), which is specifically designed to discover interesting structures in continuous features, is proposed. The MD plot does not require adjusting any parameters of density estimation, which is what may make the use of this plot compelling particularly to non-experts. The visualization tools in question are evaluated against statistical tests with regard to typical challenges of explorative distribution analysis. The results of the evaluation are presented using bimodal Gaussian, skewed distributions and several features with already published PDFs. In an exploratory data analysis of 12 features describing quarterly financial statements, when statistical testing poses a great difficulty, only the MD plots can identify the structure of their PDFs. In sum, the MD plot outperforms the above mentioned methods.

## Introduction

In exploratory distribution analysis, it is essential to investigate the structures of continuous features and to ensure that such investigations do not mislead researchers and cause making false assumptions. When given one feature in the data space, there are several approaches available to evaluating univariate structures, using indications of the quantity and range of values. These approaches include quantile-quantile plots [1, 2], histograms or cumulative density functions, and probability density functions (PDFs). When the goal is to evaluate many features simultaneously, four approaches are of particular interest: the Box-Whisker diagram (box plot) [3], the violin plot [4], the bean plot [5] and the ridgeline plot [6]. Since the box plot

**Competing interests:** The authors state hereby that they have no competing interests.

and it's counterpart the range bar [7], together with its extension, the notched box plot [8], are nearly unable to visualize multimodality [3] they are disregarded in this work. On the other hand, the violin plot, as suggested by the name, was specifically intended to identify multimodality by exposing the waist between two modes of distribution.

In exploratory statistics, univariate density estimation is a challenging task, especially to non-experts in the field. In fact, changing the default parameters of the available software, such as the bandwidth and kernel density estimator, can not only lead to better results but also to worse ones, using the abovementioned methods. Morover, both in a strictly exploratory setting and when evaluating quality measures for supervised or unsupervised machine learning methods, it is difficult to set those parameters without having a prior model of the data or results of the evaluation. Hence, nonexperts typically use the default option. On the one hand, it is a challenging task to consider the intrinsic assumptions of common density estimate approaches, which leads to opt for using the most common methods in their default setting. On the other hand, "wisely used, graphical representations can be extremely effective in making large amounts of certain kinds of numerical information rapidly available to people" [9], p. 375.

When the default parameter settings are used, the schematic plots of violin plots, bean plots, ridgeline plots and histograms provide misleading visualizations that will be illustrated for several bodies of data. Thus, it is necessary to develop a new graphical tool that enables a better understanding of the data at hand. This work proposes a strictly data-driven schematic plot, called the mirrored-density plot, based on Pareto density estimation (PDE). The PDE approach is particularly suitable for detecting structures in continuous data and in addition its kernel density estimation does not require any parameters to be set. The MD plot is compared with conventional methods, like violin plots, bean plots, ridgeline plots and histograms. This work will show that, for multimodal or skewed distributions, the MD plot is able to investigate distributions of data with more sensitivity than conventional methods. Statistical testing will be used as an indicator of the sensitivity of all the methods in terms of skewness and multimodality. For exploratory data analysis in a high-dimensional case, descriptive statistics will be used to show that the bean plot, unlike the MD plot, gives misleading visualizations.

## Methods

The methods section is divided into three parts. First, we outline how the performance of visualization tools is investigated. The focus of interest in this work lies in a separate visualization of basic properties of the empirical distribution of each feature, which means that our interest is restricted to univariate density estimation and visualizations that can present more than one feature in one plot. Such approaches are usually called schematic plots. The best-known representative is the box-whisker diagram (box plot) [3]. However, box plots are unable to visualize multimodality (e.g., [10]) and are therefore not investigated herein. In the second section, we introduce and compare the visualization tools. In the last section, we introduce the MD plot.

### Performance comparison

In this work, three steps of comparison are applied. First, artificial features are generated by taking specifically defined sampling approaches. Thus, the basic properties of the investigated distributions are well defined, as long as the sample size is not too small. For the artificial datasets in the case of skewness and bimodality, samples are chosen for their maximum size allowable for exact statistical testing. On the other hand, the minimum size is chosen for the artificial dataset of the uniform distribution for which a QQ plot against the uniform distribution would indicate a straight line. The here investigated sample sizes for natural and artificial datasets range from 269 to 31.000. The implicit assumption of this work is that with this range of sample sizes,

it is not probable that the results of the compared methods will change. In the case of the MD plot, the underlying Pareto density estimation is well-investigated for varying sample sizes [11]. To account for variance in sampling, we perform 100 iterations of sampling and test the artificial datasets for multimodality and skewness to visualize them with schematic plots.

The sensitivity for multimodality is compared with Hartigan's dip statistic [12] because it has the highest sensitivity in distinguishing unimodality from nonunimodality when compared to other approaches [13]. For skewness, the D'Agostino test of skewness [14] is used to distinguish skewed distributions from normal distributions. In the next step, natural features are selected and the basic properties of the empirical distributions are already known. The first and second steps outline the challenges the conventional methods face.

In the last step, we exploratively investigate a new dataset containing several features with unknown basic properties to summarize the problems with visualizing the estimated probability density function. In such a typical data mining setting, it would be a very challenging task to adjust the parameters of conventional visualization tools. For example, when visualizing high-dimensional data, one is unable to set the parameters of a method correctly because the appropriate adjustments are unknown (e.g., p.42, Fig 5.2 in [15]) or one sets the parameters for a specific dataset correctly because the option is known beforehand [16]. However, this option automatically becomes inappropriate for a dataset with unsimilar properties (e.g., p.8, Fig 7, p. [16] see also the example in S6 File, section 4). In sum, the right choice of parameters is interrelated with the properties of data that are unknown in an unsupervised or exploratory data mining task. Table 1 summarizes the interesting basic properties from the perspective of data mining and the methods used to compare the performance of different methods. Extensive knowledge discovery for this dataset was performed in [17]. Therefore, we compare the visualizations with basic descriptive statistics and show which visualization tools do not visualize the shapes of the PDF accurately without changing the default parameters of the investigated visualization methods.

Comparing visualizations is challenging because they have the same problems as the estimation of quantiles or clustering algorithms such as k-means or Ward: they depend on the specific implementation (c.f. [18–21]). Therefore, this work restricts the comparison to several conventional methods and specifies the programming language, package and PDF estimation approach used to outline several relevant problems for visualization of the basic properties of the PDF. To ensure that the MD plot introduced herein does not depend on a specific implementation, we use two different programming languages (R and Python), and the results from R presented herein are reproduced in the Python tutorial attached to this work.

## Visualization tools

Usually, univariate density estimation is either based on finite mixture models or variable kernel estimates or uniform kernel estimates [11]. Finite mixture models attempt to find a

**Table 1. Summary of basic properties of empirical distributions that are interesting for data mining.**

| Interesting basic Properties | Exemplary data mining applications | Statistical test used | Descriptive Statistic |
|---|---|---|---|
| Uniformity versus multimodality | Biomedical data [22], Water vapor [23] | Hartigan's dip test [12] | Difference between mean and median can indicate multimodality, several coefficients [23] |
| Data clipping versus heavy-tailedness | Flood data [24], Upper Income [25] | Not required here, but we can refer to [24, 26] | Range of data is sufficient for the task. "There is no easy characteristic for heavy-tailedness" [27] |
| Skewness versus normality | Biomedical data [28], Strength of Glass Fibers & Market Value Growth [29] | D'Agostino test [14] | Third order statistics, for example [28] |

superposition of parameterized functions, typically Gaussians, that best account for the data [30]. In the case of kernel-based approaches, the actual probability density function is estimated using local approximations [30]: the local approximations are parameterized in such a way that only data points within a certain distance of a selected point influence the shape of the kernel function, which is called the (band-)width or radius of the kernel [30]. Variable kernel methods have the capacity to adjust the radius of the kernel, and uniform kernel algorithms use a fixed global radius [30]. Histograms use a fixed global radius to define the width of a bin (binwidth). The binwidth parameter is critical for the visualized basic properties of the PDF, and in this work, only the default parameter will be used for the reason that non-experts might not adjust the parameters on their own. However, there are approaches available for a more elaborate option depending on the intrinsic assumptions about the data (e.g., [31]). As an example, we use histograms of plotly [32], which can be used in either R or MATLAB or Python. This work concentrates on visualizing the estimated probability density distribution (PDF), which will be called the distribution of the feature (variable).

The first variation visualizing the PDF was the vase plot [33], where the box of a box plot is replaced by a symmetrical display of estimated density [10]. The box plot itself visualizes only the statistical summary of a feature. A further amendment was the violin plot, which mirrors an estimated PDF so that the visualization looks similar to a box plot. "The bean plot [5] is a further enhancement that adds a rug that is showing every value and a line that shows the mean. The appearance of the plot inspires the name: the shape of the density looks like the outside of a bean pod, and the rug plot looks like the seeds within" [10].

The violin plot [4] uses a nonparametric density estimation based on a smooth kernel function with a fixed global radius [34]. The R package 'vioplot' on CRAN [35] serves as a representative for this work and uses the density estimation with the bandwidth defined by a Gaussian variance of the R package 'sm' on CRAN [36]. Another commonly applied estimation method is using the density estimation of the R package 'stats' [37], where the bandwidth is usually computed by estimating the mean integrated square error [38], nevertheless, several other approaches can be chosen as well.

An alternative to the "vioplot" is the geom_violin method [4] of the well-known "ggplot2" package [39] presented in S6 File, which uses the density estimation specified in [37]. In contrast to the violin plots, the bean plot in the R package 'beanplot' on CRAN [5] redefines the bandwidth [40]. As noted by Bowman and Azzalini, the density estimation critically depends on the choice of the width of the kernel function [34].

Yet another approach are ridgeline plots. "Ridgeline plots are partially overlapping line plots that create the impression of a mountain range" [41]. In R, they are available in the ggridges packages on CRAN [41] and either use the density estimation approaches of R discussed above (if set manually) or the default setting which "estimates the data range and bandwidth for the density estimation from the entire data at once, rather than from each individual group of data" [41]. The default setting is used in this work.

One of the most common ways to create a violin plot in Python is to use the visualization package 'seaborn' [42], which extends the Python package 'matplotlib' by statistical plots such as the violin plot. Seaborn uses Gaussian kernels for kernel density estimation from the Python package 'scipy' [43], where the bandwidth is set to Scott's Rule by default (see https://github.com/scipy/scipy/blob/v1.3.0/scipy/stats/kde.py#L43-L637) [30]. The density plots and ridgeline plots in Python, presented in supplementary E, are created by using the 'kdeplot' function of the 'seaborn' package. This approach uses the density estimation by Racine [44] implemented in the 'statsmodel' package [45] if it is installed. If it is not installed, the density estimation of 'scipy' is used.

## Mirrored Density plot (MD plot)

A special case of uniform kernel estimates is the density estimation using the number of points within a hypersphere of a fixed radius around each given data point. In this case, the number of points within a hypersphere of each data point is used for the density estimation at the center of the hypersphere. In "Pareto density estimation (PDE), the radius for hypersphere density estimation is chosen optimally [with respect to] information theoretic ideas" [11]. Information optimization calls for a radius that enables the hyperspheres to contain a maximum of information using minimal volume [11]. If a hypersphere contains approximately 20% of the data on average, it is the source of more than 80% of the possible information any subset of data can have [11]. PDE is particularly suitable for the discovery of structures in continuous data and allows for the discovery of mixtures of Gaussians [22].

For this work, the general idea of mirroring the PDF in a visualization is combined with the PDE approach to density estimation resulting in the Mirrored-Density plot (MD plot). Using the theoretical insights of [11] for the Pareto radius and [31] for the number of kernels, the PDE algorithm is implemented in the package 'DataVisualizations' on CRAN [46] and in addition independently implemented in Python [47]. To provide an easy-to-use method for non-experts, the MD plot allows for an investigation of the distributions of many features (variables) after common transformations (symmetric log, robust normalization [48], percentage) with automatic sampling in the case of large datasets and several statistical tests for normal distributions. If all tests agree that a feature is Gaussian distributed, then the plot of the feature is automatically overlaid with a normal distribution of robustly estimated mean and variance equal to the data. This step allows the marking of possible non-Gaussian distributions of single feature investigations with a quantile-quantile plot in cases where statistical testing may be insensitive. In the default mode, the features are ordered by convex, concave, unimodal, and nonunimodal "distribution shapes".

The MD plot performs no density estimation below a threshold defining the minimal amount of unique data. Instead, a 1D scatter plot (rug plot) is visualized in which for each unique value, the points are jittered on the horizontal (y-)axis to indicate the number of points per unique value. Another threshold defines the minimal amount of values in the data below which a 1D scatter plot is presented instead of a density estimation. The default setting of both thresholds can be changed or disabled by the user if necessary. These thresholds are advantageous in case of a varying amount of missing data per feature or if the benchmarking of algorithms yields quantized error states in specific cases (S6 File, section 5).

The MD plot can be applied by installing the R package 'DataVisualizations' on CRAN [46] in the framework of ggplot2 [39]. The Python implementation of the MD plot is provided in the Python package 'md_plot' on PyPi [47]. The vignettes describing the usage and providing the data are attached to this work for the two most common data science programming languages, namely, Python and R. In the next section, the visual performance indicating the correct distribution of features is investigated by a ridge line plot, a violin plot, a bean plot and a histogram and compared against an MD plot.

## Results

Initially, a random sample of 1000 points of a uniform distribution was drawn and visualized by a commonly used ridgeline, violin, bean, MD plot (Fig 1) and histogram (Fig A in S4 File) and in the corresponding methods in Python (Fig A in S3 File, Fig A in S5 File). In PDF visualizations of a uniform distribution, a straight line is expected, with possible minor fluctuations depending on the random number generator used (range [-2,2], generated with R 3.5.1, runif function). Contrary to expectations, the ridgeline plot, histogram and bean plot indicate

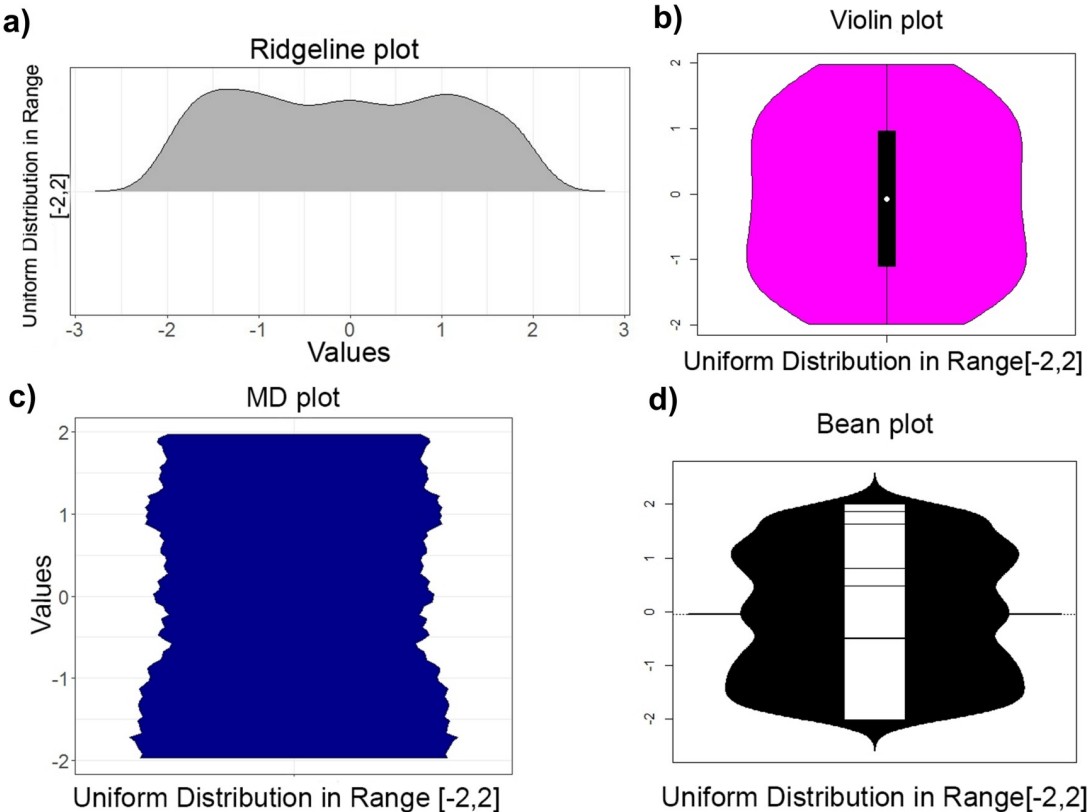

**Fig 1.** Uniform distribution in the interval [−2,2] of a 1 000 points sample visualized by a ridgeline plot (a) of ggridges on CRAN [41] (top left) and violin plot (b, top right), bottom: bean plot (d, right) and MD plot (c, left). In the ridgeline, violin and bean plot, the borders of the uniform distribution are skewed contrary to the real amount of values around the borders 2,−2. The bean plot and ridgeline plot indicate multimodality but Hartigan's dip statistic [12] disagrees: p(n = 1 000,D = 0. 01215) = 0.44.

multimodality, and the bean plot, ridgeline plot, and violin plot bend the PDF line in the direction of the end points. The visualization of this sample in Python with the package 'seaborn' [42] shows a tendency towards multimodality (Fig A in S3 File). Hartigan's dip test [12] and D'Agostino test of skewness [14] yield p(N = 1 000, D = 0.01215) = 0.44 and p(N = 1 000, z = 0.59) = 0.55, respectively, indicating that this sample is unimodal and not skewed.

As a consequence, several experiments and one exploratory investigation of a high-dimensional dataset are performed. The first two experiments investigate the multimodality and skewness of the data. The third experiment investigates the clipping of data, which is often used in data science. The fourth experiment uses a well-investigated clipped feature that is log-normal distributed and possesses several modes [49]. In the exploratory investigation, descriptive statistics in a high-dimensional case are used to outline major differences between the bean plot and the MD plot. In the last experiment, the effect of the range of values on the schematic plots is outlined.

## Experiment I: Multimodality versus unimodality

Two Gaussians, where the mean of one is changed, were used to investigate the sensitivity for bimodality in the ridgeline, violin bean, MD plot and histogram (Fig B in S4 File), as well as in Python (Fig B in S5 File). For each Gaussian randomized sample, 15 500 points were drawn,. The sample consisting of the first Gaussian N (m = 0, s = 1) remained unchanged (for

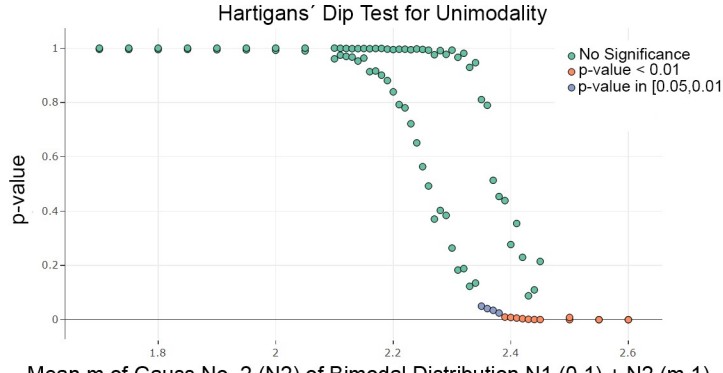

**Fig 2. Scatterplots of a Monte Carlo simulation in which samples were drawn and testing was performed in a given range of parameters in 100 iterations.** The visualization is restricted to the median and 99 percentile of the p-values for each x value. The test of Hartigan's dip statistic is highly significant for a mean higher than 2.4 in a sample of size n = 31.000.

definition of Gaussian mixtures please see [50]), and the second Gaussian N (m = i, s = 1) changed its mean through a range of values. Vividly, the distance between the two modes of a Gaussian mixture varies with each change of the mean of the second Gaussian. For statistical testing with Hartigan's dip test, 100 iterations were performed to take the variance of the random number generators and statistical method into account. Fig 2 shows that starting with a mean of 2.4, a significant p-value of approximately 0.05 is probable, and starting with a mean of 2.5, every p-value will be below 0.01.

This result is visualized in Fig 3. The bimodality is visible in the ridgeline plot and bean plot starting with a mean equal to 2.4 and in the MD plot starting with a mean equal to 2.4. However, a robustly estimated Gaussian in magenta is overlaid on the MD plot, making bimodality visible starting with a mean of 2.2. The Hartigan dip statistic [12] is consistent with these two schematic plots. In contrast, the violin plots examined here, except for geom_violin of ggplot2

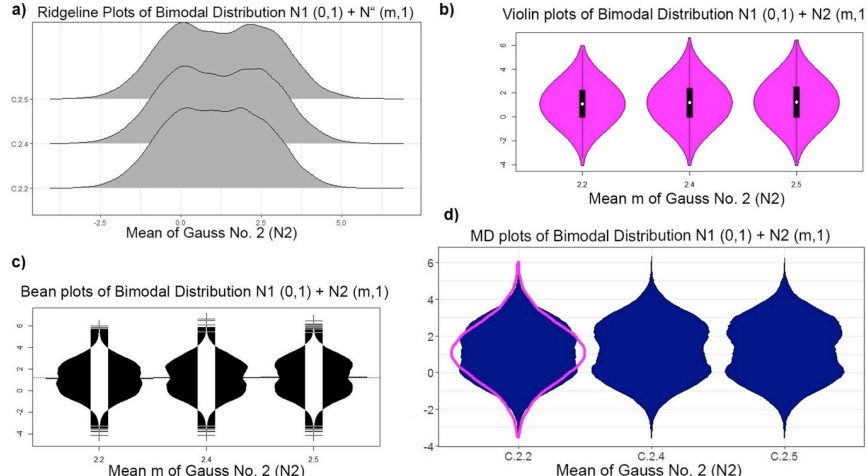

**Fig 3.** Plots of the bimodal distribution of changing mean of second Gaussian: Ridgeline plots (a) of ggridges on CRAN [41], violin plot (b), bean plot (c), and MD plot (d). Bimodality is visible beginning with a mean of 2.4 in a bean plot, ridgeline plot and MD plot, but the MD plot draws a robustly estimated Gaussian (magenta) if statistical testing is not significant, which indicates that the distributions are not unimodal with a mean of two. The bimodality of the distribution is not visible in the violin plot [4] of the implementation [34]".

(see S6 File), do not show a bimodal distribution (Fig 3), while the Python violin plots and ridgeline plots show the bimodality starting with a mean equal to 2.4 (Fig B in S3 File, Fig B in S5 File). Histograms are less sensitive, showing a bimodal distribution beginning with a mean of 2.5.

## Experiment II: Skewness versus normality

Next, an artificial feature of a skewed normal distribution is generated by the sampling method of the R package 'fGarch' available on CRAN [51]. For the skewed Gaussian, large randomized samples of 15 000 points were drawn for each value of the skewness parameter. The case of *N (m = 0,s = 1,xi = 1)* defines the uniform Gaussian distribution (for definition of Gaussian please see [50], skewed distributions [51]). One hundred iterations were performed, and the D'Agostino test of skewness [14] revealed no significant results (for skewness) in a range of [0. 95,1.05] in Fig 4. Skewness is visible in the bean plot and MD plot (Fig 5) but not in the violin plot. Unlike the R version, the skewness is visible in the Python version of the violin plot (Fig C in S3 File, Fig C in S5 File) but is slightly less sensitive than the bean plot and MD plot. In the histogram, the skewness of the distribution is difficult to recognize (Fig C in S4 File). The bean plot and MD plot are slightly less sensitive with regard to skewed distributions compared to statistical testing (Fig 4).

## Experiment III: Data clipping versus heavy-tailedness

The municipality income tax yield (MTY) of German municipalities of 2015 [46, 52] serves as an example for data clipping in which the comparison will be restricted to bean plots and MD plots. MTY is unimodal. Hartigan's dip statistic is consistent with the assessment that MTY is unimodal, $p(n = 11194, D = 0.0020678) = 0.99$, [12]. The bean plot has a major limitation for clipped data, Fig 6 shows that it estimates nonexistent distribution tails and visualizes a density above and below the range of the clipping [1800,6000]. This issue can also be observed in the Python violin and density plots (Fig D in S3 File, Fig D in S5 File).

## Experiment IV: Combining multimodality and skewness with data clipping

Here, one feature is used to compare the histogram and the schematic plots against each other. The feature is the income of German population from 2003 [49]. The whole feature was modeled with a Gaussian mixture model on the log scale and verified with the Xi-quadrat-test (p <

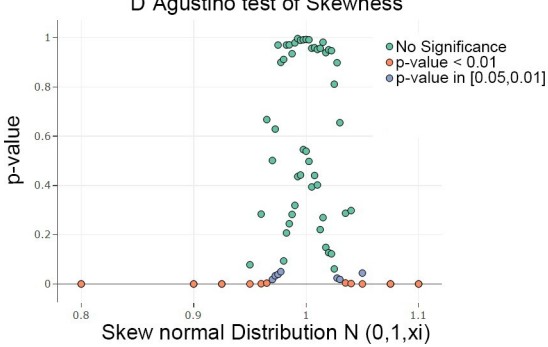

**Fig 4. Scatterplots of a Monte Carlo simulation in which samples were drawn and testing was performed in a given range of parameters in 100 iterations.** The visualization is restricted to the median and 99 percentile of the p-values for each x value. The D'Agostino test of skewness [14] was highly significant for skewness outside of the range of [0.95,1.05] in a sample of n = 15.000. Scatter plots were generated with plotly [32].

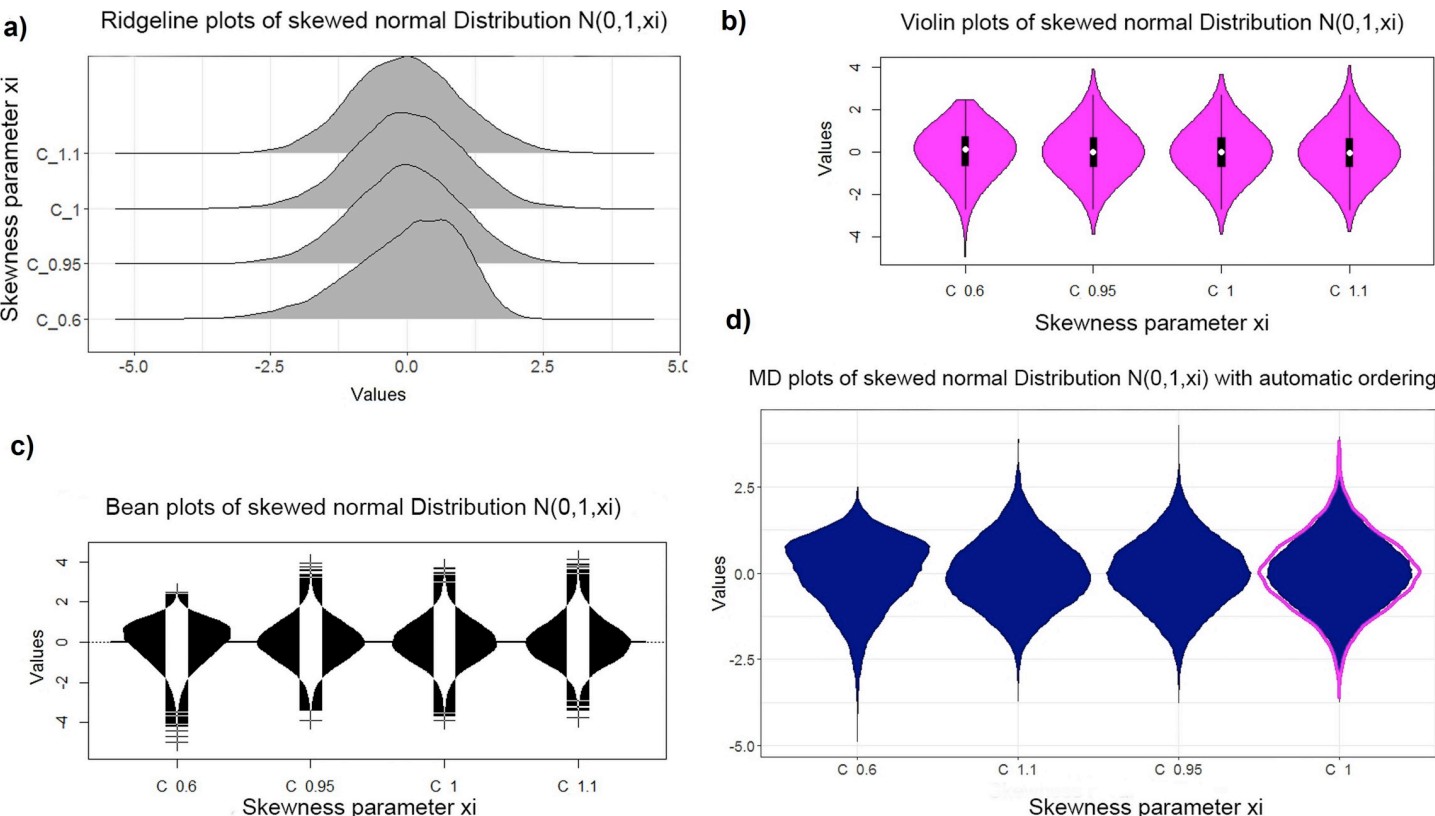

**Fig 5.** Plots of skewed normal distribution with different skewness using the R package fGarch [51] on CRAN: Ridgeline plots (a) of ggridges on CRAN [41], violin plot (b), bean plot (c) and MD plot (d). The sample size is n = 15000. The violin plot is less sensitive to the skewness of the distribution. The MD plot allows for an easier detection of skewness by ordering the columns automatically.

.001) and QQ plot [49]. A sample of 500 cases was taken and the PDF of the sample was skewed on the log scale in accordance with the D'Agostino skewness test (*skew* = -1.73, p-value p(N = 500, $z$ = -22.4) $<$ 2.2$e$ -16, [14]).

In Fig 8, it is visible that the violin plot, contrary to the MD plot, underestimates the skewness of the distribution. In addition, the violin, ridgeline and bean plots show a mode between 4 and 4.5 in the skewed distribution, (Figs 7 and 8). In Fig D of S4 File, the histogram is consistent with the MD plot and inconsistent with the bean plot, indicating that there are no values above 4.35; this means that the ridgeline and bean plot visualize a PDF above the maximum value (marked with red lines). Thus, similar to experiment III, the bean plot incorrectly visualizes a density above the maximum possible value of 4.35 with a strong tendency to underestimate it toward the maximum value, whereas the MD plot estimates density correctly (c.f. visualizations in [46]). Similar to the bean plot, the Python density function and the violin plot show values above 4.35 however they smooth the distribution more (Fig E in S3 File, Fig E in S5 File) hence these plots do not indicate multimodality.

## Experiment V: Visual exploration of distributions

The high-dimensional dataset (d = 45) of quarterly statements of companies listed on the German stock market is investigated by selecting 12 example features. It should be noted that the other features have a similar effect, but more features would make this example harder to understand. In line with the prime standard of "Deutsche Börse" [53] these companies are

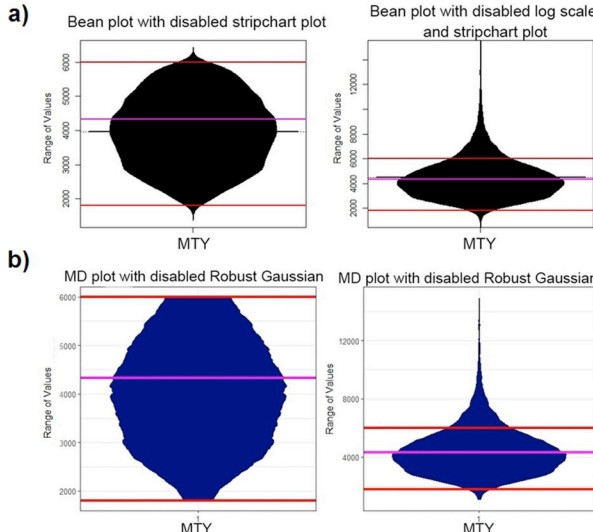

**Fig 6. MTY feature clipped in the range marked in red with a robustly estimated average of the whole data in magenta (left) and not clipped (right).** The bean plot (a) underestimates the density in the direction of the clipped range [1800, 6000] and draws a density outside of the range of values. Additionally, this leads to the misleading interpretation that the average lies at 4000 instead of 4300. The MD plot (b) visualizes the density independently of the clipping. Note that for a better comparison, we disabled the additional overlaying plots.

required to report their balance and cash flow regularly every three months in a standardized way, which are then accessible in [54]. Using web scraping, the information of n = 269 cases were extracted. In such a high-dimensional case, statistical testing, parameter settings, usual density plots and histograms become very troublesome and thus are omitted in this work. Moreover, integrating different ranges in one visualization also poses a challenge. In Table A in S2 File, the order of the descriptive statistics of the features from top to bottom is the same as in the MD plot, ridgeline plot and bean plot from left to right.(Fig 9) The MD plot enables a concave ordering, which is used here. The MD plot (Fig 9), the bean plot (Fig 10A) and the ridgeline plot (Fig 10B) visualize all features in one picture. Table A in S2 File shows that six features from right to left do not possess more than 1% negative values. Fifty percent of the data for "net tangible assets" and "total cash flow from operating activities" lie in a small positive range. "Interest expense" and "capital expenditures" do not have more than 1% positive values. "Net income" has only 25% of data below zero, and "treasury stock" has the second largest kurtosis of the selected features.

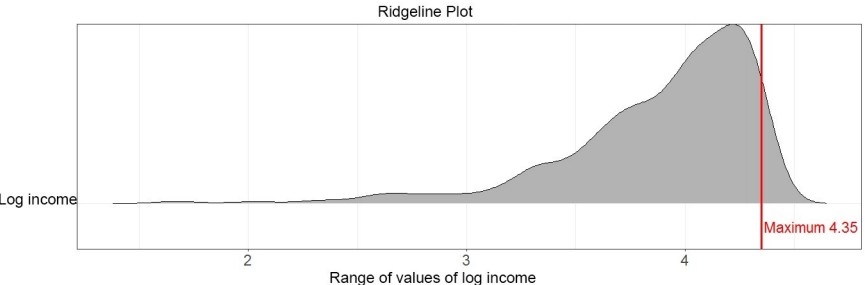

**Fig 7.** Distribution analyses performed on the log of German population's income in 2003 with ridgeline plots (a) of ggridges on CRAN (37) do not indicate clipping or multimodality.

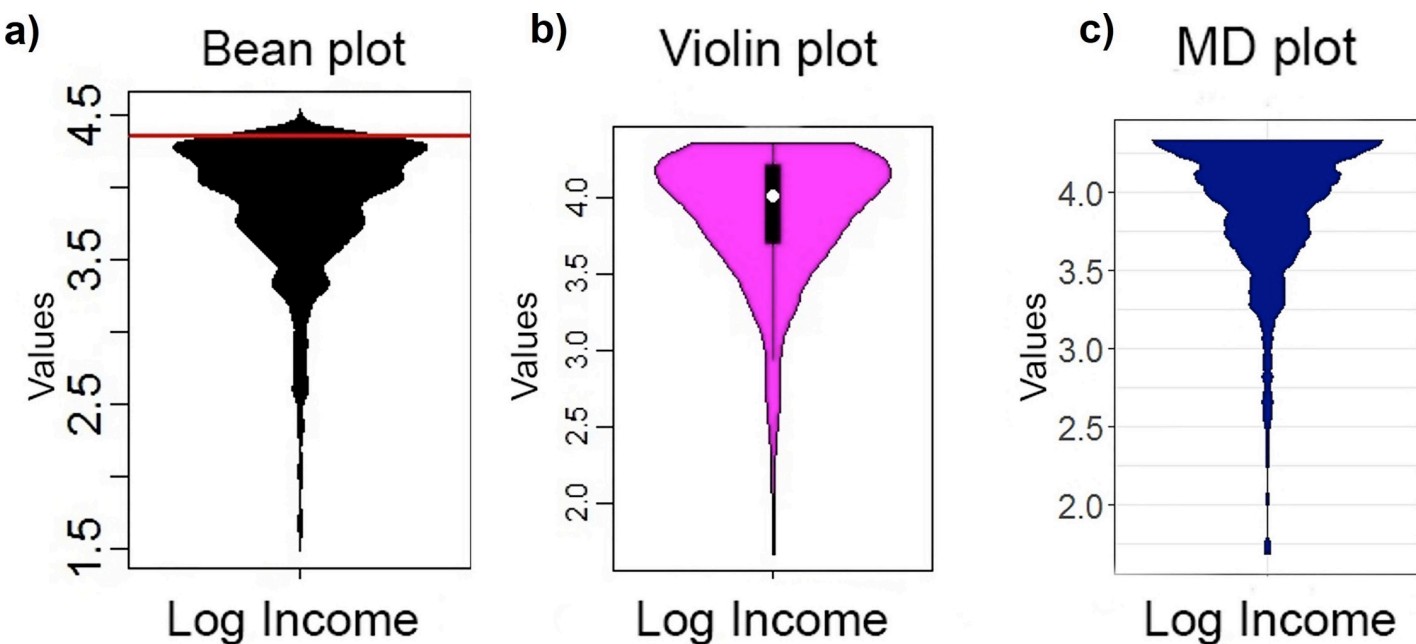

**Fig 8.** Distribution analyses performed on the log of German population's income in 2003 with the violin plot (b), bean plot (a) and MD plot (c). The bean plot and violin plot visualize an additional mode in the range of 4–4.5. The bean plot visualizes a PDF above the maximum value (red line). The multimodality of ITS is not visible with the default binwidth. Only the MD plot visualizes a clearly clipped and skewed multimodal distribution. Note that for a better comparison, we disabled the additional overlaying plots.

The MD plot shows that "net income", "treasure stock" and "total cash flow from operating activities" have a high kurtosis in a small range of data centered around zero (Fig 9). "Interest expenses" and "capital expenditures" are highly negatively skewed. The last six features from right to left do not possess visible negative values.

The bean plot changes skewed distributions into unimodal or uniform distributions (Fig 10A). In the bean plot and ridgeline plot (Fig 10B) there are no hard cuts around zero (red

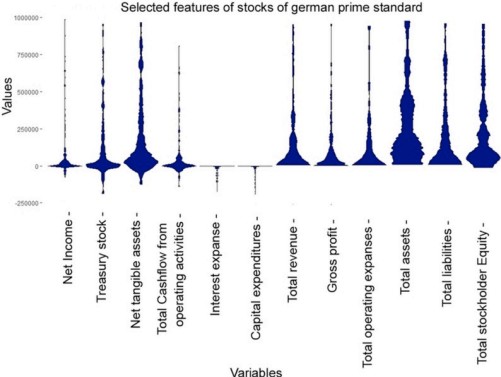

**Fig 9. MD plots of selected features from 269 companies on the German stock market reporting quarterly financial statements by the prime standard.** The features are concave ordered and the same as in Fig 10 and Table A in S2 File. For 8 out of 12 distributions, there is a hard cut at the value zero which overlaps with Table A in S2 File. The features are highly skewed besides net tangible assets, total assets, and total stockholder equity. The latter two are multimodal.

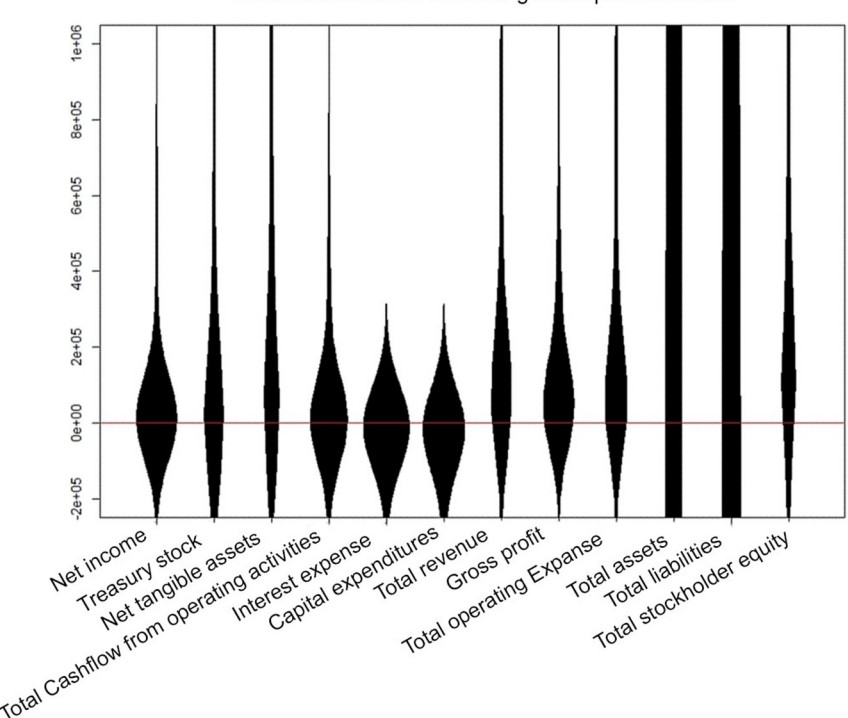

Selected features of stocks of german prime standard

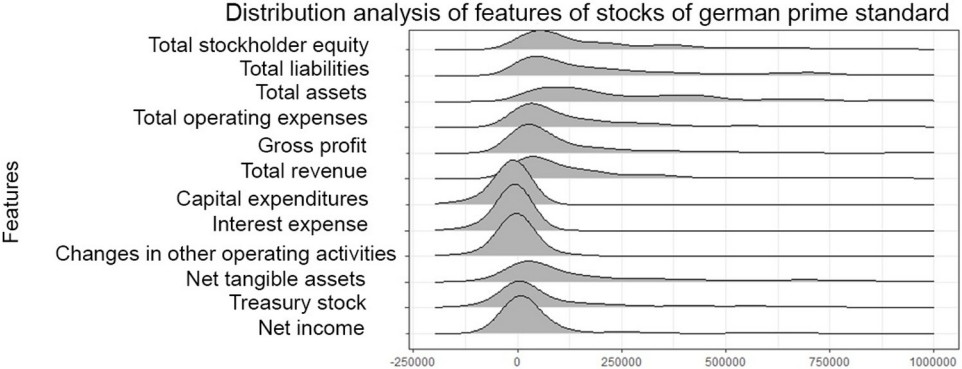

Distribution analysis of features of stocks of german prime standard

**Fig 10.** Bean plots of selected features from 269 companies on the German stock market reporting quarterly financial statements by the Prime standard (top, a) and ridgeline plots (b, bottom) of ggridges on CRAN (37). The features are concave ordered and the same as in Fig 9. There is no hard cut around the value zero (red line), and the features are unimodal or uniform with a large variance and a small skewness. The visualizations disagrees with the descriptive statistics in Table A in S2 File. Note that for a better comparison, we disabled the additional overlaying plots in bean plots.

line). Instead, approximately one-third or more of the distributions visualized lie below zero, contrary to the descriptive statistics where six features cannot have more than 1% of values below zero. In sum, the visualization of the MD plot is consistent with the descriptive statistics (Table A in S2 File) and inconsistent with the bean plot and ridgeline plot. The Python violin and ridgeline plots show values above and below the limits of [-250000, 1000000] and less detailed and incorrectly unimodal distributions (Fig F in S3 File, Fig F in S5 File).

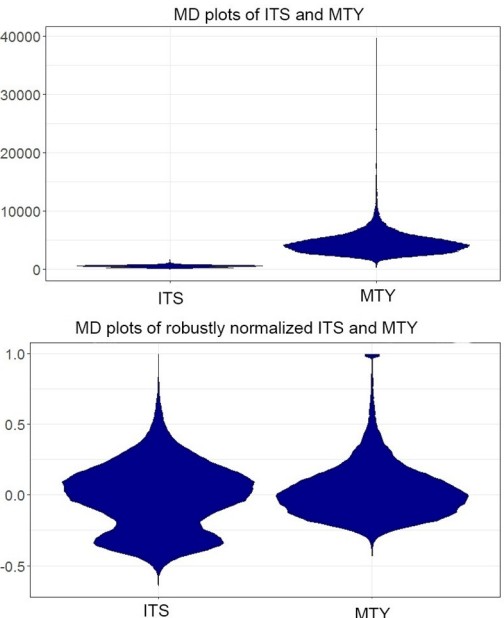

**Fig 11.** Visualization of the distribution of as few as two features at once is incorrect if the ranges vary widely (a). This is shown on the example of the MD plot (a). However, the MD plot enables the user to set simple transformations enabling the visualization of several distributions at once even if the ranges vary (b).

## Experiment VI: Range of values depending on features

In a dataset, the ranges of features often differ. For example, the range of MTY and the range of ITS (Income Tax Share, [Thrun/Ultsch, 2018; Ultsch/Behnisch, 2017]) vary widely, and the usual schematic plot would not show the distributions of both features simultaneously, which is visualized by the MD plot in Fig 11. With an option of the robust normalization [48] that is selected selected in the MD plot, the distributions can be investigated at once without changing the basic properties (Fig 11). As a result, the bimodality of the ITS feature becomes visible in the MD plot and in the bean plot (Fig A in S2 File). The violin plot, however, is unable to visualize the bimodal distribution, and the overlayed histogram underestimates it significantly (Fig E in S4 File). The Python density and violin plots draw data above and below the limits of the data but show the bimodality of the ITS feature (Fig A in S4 File, Fig G in S5 File). Statistical testing confirms that the distribution of ITS is not unimodal, $p(n = 11194, D = 0.01196) < 2.2e\text{-}16$.

## Discussion

If a simultaneous explorative distribution analysis of several features is required, the interesting basic properties of empirical distributions are depicted in Table 1: skewness, multimodality, normality, uniformity, data clipping, and the visualization of the varying ranges between features.

Usually, density estimation and visualization approaches are investigated independent of each other. Instead, the authors conflate the issue of density estimation with visualization following the perspective of Tufte, Wilk and Tukey that a graphical representation itself can be used as an instrument for reasoning about quantitative information [9, 55] (p.53). The results show that the MD plot is the only schematic plot which is appropriate for every case and

where adjustments by various parameters are not required for its process of density estimation.

Three artificial and four natural datasets show limitations of the schematic plots of ridgeline, bean, and violin plots (R and Python versions). A comparison of results with conventional statistical testing and histograms is included. The results illustrate that the usefulness of the ridgeline, violin and bean plot depends on the density estimation approach used in the algorithm, and the density approach critically depends on the bandwidth of the kernel function.

For an artificial distribution of two equal sized Gaussians and a skewed Gaussian, statistical testing was performed with the dip statistic by changing the mean of the second Gaussian and using the D'Agostino test of skewness [D'Agostino, 1970] by changing the skewness parameter (sample size n = 15000). The minimal quality requirement with regard to schematic plots is that the visualizations must at least produce comparable results to ("are as sensitive as") statistical testing and descriptive statistics. In this respect, the comparison of performance showed that the ridgeline, bean, ggplot2's violin, and MD plot have a similar sensitivity in line with statistics for bimodality and skewness as long as a sample is large enough (Figs 2–5), but for smaller sample sizes the MD plot outperforms (see Figs 1 and 9 and Table A in S2 File). The sensitivity of the Python violin plot in these cases is comparable to the sensitivity of the bean plot in R. However, overlaying the MD plot with a robustly estimated Gaussian allows for an even higher sensitivity than statistical testing. Contrary to the bean plot and the Python violin plot, the MD plot does not indicate multimodality in uniform distributions.

Automatic ordering of the features makes skewness more clearly visible in the MD plot in comparison with the ridgeline, bean, and Python violin plot. The natural example of the log of German population's income showed that for smaller samples (n = 500), the ridgeline and bean plot visualize unimodal distributions instead of skewed distributions, in contrast to the histogram and MD plots. Additionally, the ridgeline and bean plot visualize a mode that is partly above the maximum value of 4.35. The same behavior regarding stretching over the valid value range and stronger smoothing of the representation could also be observed with the Python versions. The general recommendation is that "the larger the share of graphics ink devoted to data, the better, if other relevant matters being equal [55], (p 96). Tukey and Wilk suggest avoiding undue complexity of form in summarizing and displaying [9], p. 377. Tuftte strongly argues to "erase non-data-ink within reason" [55] (p.96). Hence, the tails of violin-like schematic plots should never extend past the range of data. For clipped data, the density estimates of the MD plot do not change, contrary to the bean plot.

Kampstra proposed adding a rug (1D scatter plot) to the violin plot in the bean plot [5]. On the one hand, plotting points in a marginal distribution can easily be misleading [56] (Fig 1), and the general recommendation is that "the number of information-carrying dimensions [. . .]depicted should not exceed the number of dimensions in data" [55] (p.71). On the other hand, if only a handful of unique values are present in the data, then density estimation is inappropriate. Thus, the MD plot does not overlay the density estimation with the 1D scatter plot. Instead, it switches automatically to 1D jittered scatter plots if density estimation results in one or more Dirac delta distributions (e.g., the error rates taken from [57] in S6 File, section 5). The scatter plots are jittered, allowing for a minor indication of the amount of data having one unique value.

Violin plots in R strongly depended on specific parameter settings in order to visualize the bimodality, which was surprising. As suggested by the name, the violin plot is particularly intended to identify multimodality by exposing a waist between two modes of the distribution, since the box plot is unable to visualize it. Additionally, the R version violin plots underestimate the skewness of the distributions. It was illustrated that histograms were less sensitive in the case of bimodality because the default binwidth was not small enough. The effects found in

the ridgeline, bean and Python violin plot for skewed distributions and clipped data were outlined further in the high-dimensional case of financial statements of the companies listed on the German stock market [53]. As an example, 12 features were selected. Here, the visualizations of the ridgeline and bean plot produced an entirely misleading interpretation of the data, unlike the MD plot (cf. Table A in S2 File). The parameter settings of all plots, apart from supplementary information F, remained at default for the reason that a non-expert user would not have the capacity for changing them and an expert user would be faced with difficulties setting density estimation parameters in a solely explorative approach for each feature separately. The effects of tuning parameters are presented exemplary for the ggplot2 method geom_violin in S6 File (section 4). Certainly, many methods can be tuned to obtain a correct result for a specific distribution if prior knowledge is used. However, the example outlines that tuning parameters for one distribution results in an incorrect visualization for another distribution. Although the Python ridgeline and the violin plots use density estimators implemented in different packages, both plots show only marginally different results with the default setting.

The general performance of MD plot seems to be sufficient for data set of sizes up to $10^5$. Pareto density estimation and, subsequently, the Pareto radius has to be computed for each feature separately, which increases the computation time accordingly. Therefore, a parallel implementation of the density estimation is planned in the next iteration. Above $10^5$, Pareto density estimation becomes computationally intensive. For big data sets ($>10^5$) MD plot uses per default an appropriate subsampling method. PDE was not investigated below a sample size of 50 [11]. Thus, below this threshold, no density estimation is performed in the default setting. Instead, a 1D scatter plot with jittered points is drawn. It should be noted that the Pareto density estimation (PDE), which is used in the MD plot, is specially designed for the detection of multimodality, which could result in an overestimation of multimodality. Such an overestimation would be visible in the "roughness" of the mirrored density of a feature.

Literature suggests that schematic plots should be wider than they are tall because such shapes usually make it easier for the eye to follow from left to right [3] (p. 129). Small multiples of the type of schematic plots usually present several features with the same graphical design structure at once. Tufte suggests that "If the nature of the data suggests the shape of the graphic follow the suggestion" [55]. Therefore, in the opinion of the authors, the vertical display of box plots [3] should be favoured to the horizontal counterpart of range parts [7], and other schematic plots such as violin plots [4] should be displayed vertically.

One of the key factors of graphical integrity is to show data variation and not design variation [55]. The schematic plots investigated here are supposed to visualize such variation by density estimation. Nonsymmetric displays are more useful in the specific task of comparing pairs of distributions to each other. Although bilateral symmetry doubles the space consumed in a graphic without adding new information, redundancy can give context and order to complexity, facilitating comparisons over various parts of data [55] (p.98). The goal of the MD plot is to make it easy to compare PDFs that are often complex. Accordingly using a symmetrical display; clipping, skewness and multimodalities are more visible in data as opposed to nonsymmetrical displays if the body of the symmetric line defined by density estimation is filled out.

In sum, the results illustrate that the MD plot can outperform histograms and all other schematic plots investigated and congruent with descriptive statistics. However, following the argumentation of Tukey and Wilk [9] in p. 375, it is more difficult to absorb broad information from tables of descriptive statistics than it is to plot all features in one picture. Typically, skewness and multimodality for each feature in Table A in S2 File would have been statistically tested, leading to an even bigger table. The MD plot offers several advantages in addition to a simple density estimation of several features at once. 1D-scatter plots below a threshold proved very helpful for the benchmarking of clustering algorithms because, in several cases, the

performance evaluation yielded discrete states (see S6 File, section 5). To the knowledge of the authors, this has yet to be reported in the literature. The MD plot allows us to investigate distributions after common transformations such as robust normalization and the overlaying of distribution with robustly estimated Gaussians. The usage of transformations is often astonishingly effective [9], p. 376. For example, using the robust transformation in combination with this type of overlaying increased the sensitivity of the tendency that a dataset possesses cluster structures compared with usual statistical testing of the 1$^{st}$ principal component [58]. Wilk and Tukey argued to "plot the results of analysis as a routine matter" [9], p.380, for which the MD plot can be a useful tool. For example, ordering features by distribution shapes proved to be helpful if the performance of classifiers is evaluated by cross-validations [59]. If the advantages are combined with the ggplot2 syntax, they provide detailed error probability comparisons [60] with a high data to ink ratio (c.f. [55] (p. 96).

## Conclusion

This work indicates that the currently available density estimation approaches in R and Python can lead to major misinterpretations if the default setting is not adjusted. On the one hand, adjusting the parameters of conventional plots would require prior knowledge or statistical assumptions about the data, which is generally challenging to acquire. On the other hand, the effective laying open of the data to display the unanticipated, is a major portion of data analysis [9], p. 371. In this case of strictly exploratory data mining, we propose a parameter-free schematic plot, called the mirrored density plot. The MD plot represents the relative likelihood of a given feature (variable) taking on specific values, using the PDE approach, to estimate the PDF. PDE is slivered in kernels with a specific width. The width, and therefore the number of kernels, depends on the data. The MD plot enables the user to estimate the PDFs of many features in one visualization. Both artificial data and natural examples forming multimodal and skewed distributions were used to show that the MD plot is a good indicator in a case of bimodal as well as skewed distributions for small and large samples. All other approaches had intrinsic assumptions about the data, which in some cases led to misguiding interpretations of the basic properties. The MD plot possesses an explicit model of density estimation based on information theory and is parameter-free as defined by a data-driven kernel radius, contrary to the commonly used density estimation approaches (like bean and violin plot). Furthermore, the MD plot has the advantage of visualizing the distribution of a feature correctly in the case of data clipping and varying ranges of features. In future research, a blind survey should be conducted to investigate how well a lay person can detect all underlying structures from the MD plots alone. In sum, the MD plot enables non-experts to easily apply explorative data mining by estimating the basic properties of the PDFs (distributions) of many features in one visualization when setting several parameters is difficult.

Combining the MD plot with a(n) (un-)supervised index is an excellent approach to evaluating the stability of stochastic clustering algorithms (e.g., [15]) or classifiers. Furthermore, it can be used with quality measures for dimensionality reduction methods to compare projection methods (e.g., [15]). The MD plot is integrated into the R package 'DataVisualizations' on CRAN [46] in the framework of ggplot2 and in the Python package 'md_plot' on PyPi [47].

## Supporting information

**S1 File. ITS and MTY.**
(DOCX)

**S2 File. Descriptive statistics.**
(DOCX)

**S3 File. Conventional violin plot in Python.**
(DOCX)

**S4 File. Overlayed histograms.**
(DOCX)

**S5 File. Density and ridgeline plots in Python.**
(DOCX)

**S6 File. Violin plot of ggplot2.**
(PDF)

**S7 File.**
(DOCX)

## Acknowledgments

We thank Felix Pape for the first implementation of the MD plot in the R package 'DataVisualizations' and Hamza Tayyab for programming the web scraping algorithm that was used to extract the quarterly statements. Special thanks go to Monika Sikora for language revision of this article and Martin Thrun for figure post-processing.

## Author Contributions

**Conceptualization:** Michael C. Thrun.

**Data curation:** Michael C. Thrun.

**Formal analysis:** Michael C. Thrun.

**Investigation:** Michael C. Thrun.

**Methodology:** Michael C. Thrun, Alfred Ultsch.

**Project administration:** Alfred Ultsch.

**Software:** Michael C. Thrun, Tino Gehlert.

**Supervision:** Alfred Ultsch.

**Validation:** Tino Gehlert.

**Visualization:** Michael C. Thrun, Tino Gehlert.

**Writing – original draft:** Michael C. Thrun.

**Writing – review & editing:** Michael C. Thrun, Tino Gehlert, Alfred Ultsch.

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
