## [Decision Letter · Decision Letter 0]

16 Dec 2019

PONE-D-19-19081

Analyzing the Fine Structure of Distributions

PLOS ONE

Dear Dr. rer. nat. Thrun,

Thank you for submitting your manuscript to PLOS ONE. After careful consideration, we have decided that your manuscript does not meet our criteria for publication and must therefore be rejected.

I am sorry that we cannot be more positive on this occasion, but hope that you appreciate the reasons for this decision.

Yours sincerely,

Qichun Zhang, PhD

Academic Editor

PLOS ONE

Additional Editor Comments (if provided):

Two reviewers returned the critical comments focusing on the novelty of the manuscript. Basically, the author redo some existing method using Python where the new features have not been demonstrated clearly. In addition, the English writing should be pre-checked where some typos would affect the readability of the manuscript.

Reviewers' comments:

Reviewer's Responses to Questions

**Comments to the Author**

1. Is the manuscript technically sound, and do the data support the conclusions?

Reviewer #1: Yes

Reviewer #2: Partly

2. Has the statistical analysis been performed appropriately and rigorously? 

Reviewer #1: I Don't Know

Reviewer #2: No

3. Have the authors made all data underlying the findings in their manuscript fully available?

Reviewer #1: Yes

Reviewer #2: Yes

4. Is the manuscript presented in an intelligible fashion and written in standard English?

Reviewer #1: Yes

Reviewer #2: Yes

5. Review Comments to the Author

Reviewer #1: The paper draws important attention to the pitfalls of existing distributional visualizations for effectively summarizing the nuances of non-normal distributions. Particularly, the paper assesses the efficacy of existing graphical representations (violin plot, box plot, bean plot) for summarizing skewed, multimodal, and uniform distributions, and provides a context and implementation to introduce 'mirror density plots' as an alternative.

The context given for existing visual tools is sound, though there are some areas that could be improved:

- Regarding histograms in section 2.2: “…in this work, only default parameter will be used because layman would probably not adjust parameters”. Given that the target user of a statistical visualization package in R or Python likely has experience in data science or statistics, this assumption warrants re-examination. For example: adding (bins=“auto”) is a common procedure for researchers using matplotlib’s built-in histogram function (see: https://docs.scipy.org/doc/numpy/reference/generated/numpy.histogram_bin_edges.html#numpy.histogram_bin_edges). Additional context for histograms could be improved with an acknowledgement of non-uniform binning methods, though this may not be in scope for the paper.

- It is surprising that this paper contains no ordinary density plots accompanied by sub-axis rugs, which are common methods for analyzing distributions. Similarly, ridgeline plots do not make an appearance, either in discussions of existing visualization methods, or in schematic comparisons of multiple dimensions. Given that these graphical representations seem more common than bean plots, for example, which are discussed at length, background context would benefit from the inclusion of more ordinary non-symmetric representations of distributions.

- The paper would benefit from a ridgeline plot with a single axis, comparing the same non-normal distribution with various binning methods (e.g. each: default histogram using n=10; SciPy's "auto" method mentioned above; Scott's Rule mentioned in paper and detailed on SciPy link above; proposed PDE method; any others potentially relevant according to authors' literature review)

The fundamental scientific contribution of the paper is the usage of the Pareto Density Estimation (PDE) to construct a visualization of a univariate distribution which captures non-normal characteristics of distributions, such as skew, multimodality, and uniformity. This method appears well-supported, and is explained concisely, with easily accessible packages for both R and Python to supplement the work.

While the PDE method for binning appears well-defended, the implementation into a visual language leaves some questions unanswered:

- In the broader data visualization community, "Mirror Density" plots are bivariate distributions: for example, one might construct two violin plots of distributions, conditioned on a second binary value (e.g. control vs. experiment), split the resultant forms in half lengthwise, and position them opposite one another to create a comparative representation of the conditioned distributions (see: https://www.d3-graph-gallery.com/graph/density_mirror.html). In this bivariate application, the comparative symmetry adds value to the analytic process. It is unclear from the paper whether the authors are aware of this namespace convergence, but independent of nomenclature, the paper would benefit from an assessment of the analytic value for making a univariate density plot symmetric.

- In section 3.5 "The high-dimensional data set (d=45)... is investigated by selecting 12 features": Ridgeline plots with the PDE binning method may be a more space-conservative method of implementing the algorithm (though admittedly, d=45 remains a non-trivial 'curse of dimensionality').

- With regard to the German stock market data in section 3.5, the schematic MD (Fig. 9) and violin (Fig. 10) plots compare distributions in very different ranges. The paper would benefit from the removal of 'InterestExpense' and 'CapitalExpenditures' from the exemplary features, perhaps to be replaced with features of a range more similar to the other features in the schematic plots.

- In general, plots should be ordered where possible, e.g. Fig. 5b,c,d should show skew parameter xi in order [0.6,0.95,1,1.1] for clarity.

- Stacked histograms are not advisable for this application. Stacked histograms make sense when considering how categories sum to a total population (e.g. when exploring various revenue sources and the resultant aggregate revenue in a single graphic). For comparing model distributions of various skewness parameters, e.g. in Fig. 3a, 5a (are these the same histogram?), stacking does not seem appropriate. Neither does stacking seem appropriate for histograms of normalized data, e.g. Robustly Normalized values for Income Tax Share (ITS) and Municipality Income Tax Yield (MTY) in Fig. 12a. The sum total of 2 bins normalized from different ranges provides no substantial comparative analytic value. In both stacked histogram cases: overlaying, rather than stacking, may provide the intended visual effect, and would be appropriate to the data.

As a reviewer with experience in data visualization, I feel confident in my assessment of this component of the paper; however, it is my hope that fellow peer reviewers can speak in a more informed manner on the statistical evaluations and experiments performed.

The paper's conclusion in Section 5 "current density estimation approaches can lead to major misinterpretations if the default setting is not adjusted" seems to suggest that the scientific community would benefit greatly from the addition of PDE binning methods to existing open-source visualization packages such as ggplot, matplotlib, seaborn and plotly. I hope the authors consider integrated contribution to existing open-source tools.

The paper would be improved by a spelling check, and a grammar proofread by a native English speaker. Typos and grammatical oversights do not obstruct communication, but do inhibit narrative flow.

Reviewer #2: This paper studies the mirrored desity (MD) plot and show various structures of the MD results. This paper proposed a MD plot implemented in Python. Since mirrorred density (MD) plot has been developed in R already, the contribution of this paper is not clearly justified. It is not clear what kind of new features this paper introduce into the MD plot.

6. PLOS authors have the option to publish the peer review history of their article (what does this mean?). If published, this will include your full peer review and any attached files.

Reviewer #1: Yes: Jane Lydia Adams

Reviewer #2: No

---

## [Author Response · Author response to Decision Letter 0]

9 Jan 2020

Dear Editor-in-Chief, Dr. Joerg Heber, Dear Editorial Board,

We are very grateful for the important suggestions written by the first reviewer. However, we are shocked by the quality of the review of the second reviewer and the oversight of the handling editor. This is the first time I ever heard of or read a review in a high-class journal with evidently false statements and no scientific contribution at all. That is not proper handling of a manuscript or reviewing. Due to the anonymity of the second reviewer, we assume competing interests. We hereby report scientific misconduct.

We have addressed false statements as detailed out in the following (our responses below the reviewers’ comments in red letters). We address the positive and helpful scientific review of the first reviewer shortly afterwards. The claims that the first reviewer has written a positive review and the second reviewer either did not read the manuscript or had competing interests become evident.

Since 20.12.19, the manuscript was reinstated and a revision was allowed. Therefore, we take the opportunity to revise the manuscript in accordance with the suggestions of the first reviewer. 

Yours Sincerely

Michael Thrun, Tino Gehlert, and Alfred Ultsch

General Comments:

Comments to the Author  1. Is the manuscript technically sound, and do the data support the conclusions?  The manuscript must describe a technically sound piece of scientific research with data that supports the conclusions. Experiments must have been conducted rigorously, with appropriate controls, replication, and sample sizes. The conclusions must be drawn appropriately based on the data presented. 

Reviewer #1: Yes

Reviewer #2: Partly

2. Has the statistical analysis been performed appropriately and rigorously? 

Reviewer #1: I Don't Know

Reviewer #2: No

3. Have the authors made all data underlying the findings in their manuscript fully available?  The PLOS Data policy requires authors to make all data underlying the findings described in their manuscript fully available without restriction, with rare exception (please refer to the Data Availability Statement in the manuscript PDF file). The data should be provided as part of the manuscript or its supporting information, or deposited to a public repository. For example, in addition to summary statistics, the data points behind means, medians and variance measures should be available. If there are restrictions on publicly sharing data—e.g. participant privacy or use of data from a third party—those must be specified.

Reviewer #1: Yes

Reviewer #2: Yes

4. Is the manuscript presented in an intelligible fashion and written in standard English?  PLOS ONE does not copyedit accepted manuscripts, so the language in submitted articles must be clear, correct, and unambiguous. Any typographical or grammatical errors should be corrected at revision, so please note any specific errors here.

Reviewer #1: Yes

Reviewer #2: Yes

Reviews

Reviewer #2: “This paper studies the mirrored desity (MD) plot and show various structures of the MD results.” 

This statement is incorrect. We propose a method for visualization of univariate probability density functions (pdf’s) in data consisting of several features to present the pdf’s in a single visualization. The results show the comparison to the state of the art methods. E.g., this was stated in the abstract of the manuscript:

“Data visualization tools should deliver a sensitive picture of the univariate probability density distribution (PDF) for each feature. Visualization tools for PDFs are typically kernel density estimates and range from the classical histogram to modern tools like bean or violin plots. These visualization tools are evaluated in comparison to statistical tests for the typical challenges of explorative distribution analysis. Conventional methods have difficulties in visualizing the pdf in case of uniform, multimodal, skewed and clipped data if density estimation parameters remain in a default setting. As a consequence, a new visualization tool called Mirrored Density plot (MD plot) is proposed which is particularly designed to discover interesting structures in continuous features”

We also motivated the usage by:

“We compare the visualizations to basic descriptive statistic and show which visualization tools do not visualize the shapes of the pdf accurately. Table 1 summarizes the interesting basic properties from the perspective of data mining and the methods used to compare performance.

Table 1: Summary of basic properties of empirical distributions that are interesting for data mining.

Interesting basic Properties Exemplary data mining applications Statistical test used Descriptive Statistic

Uniformity versus multimodality Biomedical data (13)

Water vapour (14)

Hartigans’ dip test (6)

Difference between mean and median can indicate multimodality, several coefficients (14)

Data clipping versus Heavy-tailedness Flood data (15), Upper Income (16).

Not required here, but we can refer to (15, 17)

Range of data is sufficient for the task. “There is no easy characteristic for heavy-tailedness” (18)

skewness versus normality Biomedical data (19), Strength of Glass Fibres & Market Value Growth (20)

D'Agostino test(8)

Third order statistics, e.g. (19)

”

We added in Table 1 Water vapor as further application and exemplary coefficients for bimodality.

This paper proposed a MD plot implemented in Python. 

This statement is false. We propose a method which is implemented in R and Python and compare this method with state-of-the-art methods in R and Python. We provide in R a vignette and in Python an introductory tutorial:

Examplary stated in the methods section of the manuscript in 2.1:

“To make sure that the here introduced MD plot does not depend on the specific implementation we provide the package two different programming languages (R and python) reproducing the R results presented in this manuscript below in the python tutorial attached to this work.”

Examplary the reviewed manuscript had the sentences in the methods section 2.3:

“The MD plot can be applied by using the R package ‘DataVisualizations’ on CRAN (21). In the next section, the visual performance of indicating the correct distribution of features is investigated by the histogram, violin, and bean plot in comparison to MD plot. The Python implementation of the MD plot is provided in the Python package ‘md_plot’ on PyPi (22). The vignettes describing the usage and providing the data are attached to this work for the two most-common data science programming languages Pythongand R.

And in the conclusion section:

“The MD plot is available in the R-package ‘DataVisualizations’ on CRAN (21) and in the

Python package ‘md_plot’ on PyPi (22).”

21. Thrun MC, Ultsch A. Effects of the payout system of income taxes to municipalities in Germany.

In: Papież M, Śmiech S, editors. 12th Professor Aleksander Zelias International Conference on

Modelling and Forecasting of Socio-Economic Phenomena; Cracow, Poland: Cracow: Foundation

of the Cracow University of Economics; 2018. p. 533-42.

22. Gehlert T. md_plot: A Python Package for Analyzing the Fine Structure of Distributions. 2019.

Python package.“

Using the insights of the reviewer below, we specified the sentence to

“The MD plot is integrated into the R-package ‘DataVisualizations’ on CRAN (37) in the framework of ggplot2, and in the Python package ‘md_plot’ on PyPi (38)“

Since mirrorred density (MD) plot has been developed in R already, the contribution of this paper is not clearly justified. 

This statement is false. One important goal is to show that our algorithm works indepently to the programming language and specific implementation. The authors would like to refer to 2.1:

“Comparing visualizations is challenging because they have the same issues as the estimation of quantiles or clustering algorithms like k-means or Ward: they depend on the specific implementation (c.f. (9), (10),(11, 12)). Therefore, this work restricts the comparison to several conventional methods and specifies the programming language, package and pdf estimation approach used in order to outline several relevant problems for visualization the basic properties of the pdf. To make sure that the here introduced MD plot does not depend on the specific implementation we provide the package two different programming languages (R and python) reproducing the R results presented in this manuscript below in the python tutorial attached to this work.” 

The method MD plot was developed for this manuscript by the first author. It is included for technical reasons in the package ‘DataVisualizations’ which has initially been development by the first author and others for various conventional visualization shortcuts and approaches used in Data Science. As every R package on CRAN requires a citation, this package itself is cited with a conference publication about the visual investigation of correlation coefficients. A short look into the description file on CRAN would yield

“Gives access to data visualisation methods that are relevant from the data scientist's point of view. The flagship idea of 'DataVisualizations' is the mirrored density plot (MD-plot) for either classified or non-classified multivariate data presented in Thrun et al. (2019) <arXiv:1908.06081>. ”

which clearly references this publication with a preprint published in arXiv as suggested by PLOS ONE. For any user of packages in R or python (i.e. all data scientists) this is obvious.

It is not clear what kind of new features this paper introduce into the MD plot.

As the reviewer evidently did not even read the abstract of the mansucript, it is indisputable that the features are unclear for the reviewer.

Reviewer #1: The paper draws important attention to the pitfalls of existing distributional visualizations for effectively summarizing the nuances of non-normal distributions. Particularly, the paper assesses the efficacy of existing graphical representations (violin plot, box plot, bean plot) for summarizing skewed, multimodal, and uniform distributions, and provides a context and implementation to introduce 'mirror density plots' as an alternative.

The context given for existing visual tools is sound, though there are some areas that could be improved:

We thank the first reviewer for the clear stating that our methodology is sound and problems shown in the paper important.

- Regarding histograms in section 2.2: “…in this work, only default parameter will be used because layman would probably not adjust parameters”. Given that the target user of a statistical visualization package in R or Python likely has experience in data science or statistics, this assumption warrants re-examination. For example: adding (bins=“auto”) is a common procedure for researchers using matplotlib’s built-in histogram function (see: https://docs.scipy.org/doc/numpy/reference/generated/numpy.histogram_bin_edges.html#numpy.histogram_bin_edges). Additional context for histograms could be improved with an acknowledgement of non-uniform binning methods, though this may not be in scope for the paper.

We agree with the reviewer that parameters of binning are not the scope of this paper because we already provide 30 figures in this manuscript. 

Additionally, besides the “auto” parameter in Python, the setting of parameters for the binning of histograms in any exploratory analysis is unfeasible as we stated in the manuscript: 

“In the last step, we exploratively investigate a new data set with several features with unknown basic properties in order to summarize the challenges of visualizing the estimated probability density function. In such a typical data mining setting, it would be a very challenging task to adjust parameters of the conventional visualizations tools investigated here. We compare the visualizations to basic descriptive statistics and show which visualization tools do not visualize the shapes of the pdf accurately.”

With the suggestion of the reviewer to use ridge line plots, we move the histograms into the supplementary D and instead use ridge lines plots in the manuscript itself.

- It is surprising that this paper contains no ordinary density plots accompanied by sub-axis rugs, which are common methods for analyzing distributions. Similarly, ridgeline plots do not make an appearance, either in discussions of existing visualization methods, or in schematic comparisons of multiple dimensions. Given that these graphical representations seem more common than bean plots, for example, which are discussed at length, background context would benefit from the inclusion of more ordinary non-symmetric representations of distributions.

We thank the reviewer for the suggestion to include ridgelines plots consistently throughout the paper instead of histograms using the ggridges R packages on CRAN, and the ‘kdeplot’ function of the ‘seaborn’ package in supplementary E.

We changed the abstract from

“Visualization tools for PDFs are typically kernel density estimates and range from the classical histogram to modern tools like bean or violin plots.”

To

“Visualization tools for PDFs are typically kernel density estimates and range from the classical histogram to modern tools like ridgeline plots, bean or violin plots.”

We added to the methods section in the case of R

“Another approach consists of ridgeline plots. “Ridgeline plots are partially overlapping line plots that create the impression of a mountain range” (31). They are in R available in the ggridges packages on CRAN (31) and either use the density estimation approaches of R discussed above (if set manually) or per default “estimates the data range and bandwidth for the density estimation from the entire data at once, rather than from each individual group of data” (31). The default setting is used in this work.”

And in the case of Python

“The density plots and ridgeline plots in Python presented in supplementary E are created by using the ‘kdeplot’ function of the ‘seaborn’ package. This approach uses the density estimation by Racine (34) implemented in the ‘statsmodel’ package (35) if it is installed. If it is not installed, the density estimation of ‘scipy’ is used.

“

We added to the results section the parts regarding the ridgeline plot in R and Python and the changed references with regards to the histogram in plotly. For a better overview, we marked these minor changes in this letter using Microsoft Word’s review modus:

“Initially, a random sample of 1000 points of a uniform distribution was drawn and visualized by a commonly used histogram methodridgeline plot, violin plot, bean plot, and MD plot (Fig. 1) and histogram (SI D, Fig.19) as well as for density estimation in python (SI C, Fig.13, SI E Fig. 24).”

“Fig. 1: Uniform distribution in the interval [-2,2] of a 1000 points sample visualized by a histogram ridgeline plot (a) of plotly ggridges on CRAN (32) with a default binwidth (top) of plotly and bottom: violin plot (b, left), bean plot (c, middle) and MD plot (d, right). In the ridgeline plot violin plot and bean plot, the borders of the uniform distribution are skewed contrary to the real amount of values around the borders 2,-2. The bean plot and ridgeline plot indicate multimodality but Hartigans’ dip statistic (6) disagrees: p(n=1000,D = 0. 01215)= 0.44.”

“Contrary to the expectation, the ridgeline plot, histogram and bean plot indicate multimodality and bean plot, ridgeline plot, and violin plot bend the pdf line in the direction of the end points.”

“Contrary to the expectation, the ridgeline plot, histogram and bean plot indicate multimodality and bean plot, ridgeline plot, and violin plot bend the pdf line in the direction of the end points.”

“This result is visualized in Fig. 3. The bimodality is visible in the ridgeline plot and bean plot starting with a mean equal to 2.4, and in the MD plot starting with a mean equal to 2.4. However, but a robustly estimated Gaussian in magenta is overlaid in the MD plot making bimodality visible starting from a mean of 2.2. Hartigans' dip statistic (6) agrees with these two schematic plots. In contrast, violin plots do not show a bimodal distribution (Fig. 3), while the Python violin and ridgeline plots shows the bimodality starting with a mean equal to 2.4 (SI. C, Fig. 14, SI E, Fig. 25).”

“Fig. 3: Plots of bimodal distribution of changing mean of second Gaussian: Stacked histogramRidgeline plots (a) of ggridges on CRAN (32), violin plot (b), bean plot (c), and MD plot (d). Bimodality is visible beginning from mean 2.4 in bean lot ridgeline plots and MD plot, but the MD plot draws a robustly estimated Gaussian (magenta) if statistical testing is not significant which indicates in mean of two that the distributions is not unimodal. The bimodality of the distribution is not visible in the violin plot”

“Unlike the R version, the skewness is visible in the Python version of the violin plot (SI. C, Fig. 15, SI E, Fig. 26), but slightly less sensitive as the bean plot and MD plot. In the histogram,, the skewness of the distribution is difficult to recognize ((SI D, Fig 21),Fig. 5).”

“Fig. 5: Plots of skewed normal distribution by changing the skewness using the R package fGarch (43) on CRAN: Stacked histogramRidgeline plots (a) of ggridges on CRAN (32), violin plot (b), bean plot (c) and MD plot (d). The sample is with n=15000 large. The histogram and violin plot is less sensitive for the skewness of the distribution. MD plot allows for an easier detection of skewness by ordering the columns automatically.”

“This issue can also be observed with the Python violin and density plots (SI. C, Fig. 16, SI. E, Fig. 27).”

“In Fig 8, it is visible that the violin plot underestimate skewness of the distribution contrary to the MD plot. The ridgeline, violin and bean plot shows a mode in the skewed distribution between 4 to 4.5 contrary to the MD plot (Fig. 8). In Fig. 7SI D, Fig. 22, the histogram agrees with the MD plot and disagrees with the bean plot that there are no values above 4.35 meaning that the ridgeline and bean plot visualizes a pdf above the maximum value (marked with red lines”

“The Python density and violin plot shows, like the bean plot, values above 4.35, but smoothes the distrubition more (SI. C, Fig. 17, SI E, Fig. 28) and, hence, does not indicate multimodality.”

“Fig. 7: Distribution Analyses performed on the log of German people’s income in 2003 with Ridgeline plots (a) of ggridges on CRAN (37) does not indicate clipping or multimodality. a histogram of plotly (24) with a default binwidth”

“In such a high-dimensional case, statistical testing, parameter settings, usual density plots. and histograms become very troublesome and thus are omitted in this work. Moreover, it becomes challenging to integrate different ranges in one visualization. In Tab. 1, SI B, the ordering of the descriptive statistics is from top to bottom is the same as in MD plot, ridgeline and bean plot from left to right. The MD plot enables ordering by concavity which is used here. The MD plot (Fig 9), and the bean plot (Fig. 10a) and the ridgeline plot (Fig 10b) visualizes all variables in one picture.”

“The bean plot changes skewed distributions to distributions with one mode or uniform distributions (Fig. 10a). In the bean plot and ridgeline plot (Fig. 10b) tThere is are no hard cut around the value zero (red line).”

“In sum, the visualization of MD plot is in agreement of descriptive statistics (SI B, S1 Table) and disagreement with the bean plot and ridgeline plot. The Python violin and ridgeline plots is showing values above and below the limits of [-250000, 1000000] and a less detailed, and incorrectly unimodal distributions. (SI. C, Fig. 18, SI E, Fig. 29).”

“Fig. 10: Bean plots of selected features from 269 companies on the German stock market reporting quarterly financial statements by the Prime standard (top, a) and Ridgeline plots (b, bottom) of ggridges on CRAN (37). The ordering of the features is by concavity and the same as in Fig. 9. There is no hard cut around the value zero (red line) and the variables are unimodal or uniform with a large variance and a small skewness. This The visualizations disagrees with the descriptive statistics in SI B, S1 Table. Note, that for a better comparison we disabled the additional overlaying plots in beanplots.”

“However, the violin plot is unable to visualize the bimodal distribution and the stackedoverlayed histogram underestimates it significantly (SI. DB, Fig. 1223). The Python density and violin plots draws data above and below the limits of the data, but is showing the bimodality of the ITS feature (SI. C, Fig. 19, SI E, Fig. 30).”

In the discussion: 

“Three artificial and four natural datasets show the limitations of the schematic plots of ridgeline plot, bean plot, violin plot (R and Python versions). A comparison of results to conventional statistical testing and histograms is included. The results illustrate that the usefulness of the ridgeline plot, violin or so-called bean plot depends on the density estimation approach used in the algorithm, Three artificial and four natural datasets show the limitations of the schematic plots of ridgeline plot, bean plot, violin plot (R and Python versions). A comparison of results to conventional statistical testing and histograms is included. The results illustrate that the usefulness of the ridgeline plot, violin or so-called bean plot depends on the density estimation approach used in the algorithm,…”

“…by changing the skewness parameter (sample size n=15000). Statistical testing, the ridgeline plot, bean plot and MD plot have a similar sensitivity regarding bimodality and skewness as long as the sample is large enough.”

“Automatically ordering the features makes skewness more clearly visible in the MD plot in comparison to the ridgeline plot, bean plot and Python violin plot. The natural example of the Log of German peoples income showed that for smaller samples (n=500) the ridgeline plot, bean plot visualizes unimodal distributions instead of skewed distributions disagreeing with the histogram and MD plot. Additionally, the ridgeline plot, bean plot….”

“with the Python violin plotversions. For clipped data, the density estimates of the MD plot does not change contrary to the bean plot.”

“The parameter settings of all plots until the last experiment remained at default because a non-expert user would not change them and an expert user would have difficulties to set density estimation parameters in a solely explorative approach for each feature seperatly. Although the Python ridgeline and the violin plot use density estimators implemented in different packages, both only show marginally different results with the default setting.

”

- The paper would benefit from a ridgeline plot with a single axis, comparing the same non-normal distribution with various binning methods (e.g. each: default histogram using n=10; SciPy's "auto" method mentioned above; Scott's Rule mentioned in paper and detailed on SciPy link above; proposed PDE method; any others potentially relevant according to authors' literature review)

In supplementary E we now state:

“This section covers density plots and ridgeline plots created by using the ‘kdeplot’ function of the ‘seaborn’ package. The default value (Scott's rule of thumb) of the bandwidth parameter was used.”

We want to remark that in exploratory data analysis, the setting of any parameters would become infeasible because the real pdf is unknown, and the parameters would have to be set for each pdf estimation seperatly which cannot be done for many features manually in a limited time. 

This is specified in the manuscript in the discussion:

From

The parameter settings of all plots until the last experiment remained at default because a non-expert user would not change them and an expert user would have difficulties to set density estimation parameters in a solely explorative approach”

To

“The parameter settings of all plots until the last experiment remained at default because a non-expert user would not change them and an expert user would have difficulties to set density estimation parameters in a solely explorative approach for each feature seperatly.”

And in the conclusion it was stated in the manuscript already:

“Adjusting the parameters of conventional plots would require prior knowledge or statistical assumptions about the data, which are often difficult to acquire.”

Additonally, investigating various parameters would overload the manuscript as we alreardy provide 30 figures.

The fundamental scientific contribution of the paper is the usage of the Pareto Density Estimation (PDE) to construct a visualization of a univariate distribution which captures non-normal characteristics of distributions, such as skew, multimodality, and uniformity. This method appears well-supported, and is explained concisely, with easily accessible packages for both R and Python to supplement the work.

Thank you for getting the concept of our paper and understanding the usual references of the packages.

While the PDE method for binning appears well-defended, the implementation into a visual language leaves some questions unanswered:

- In the broader data visualization community, "Mirror Density" plots are bivariate distributions: for example, one might construct two violin plots of distributions, conditioned on a second binary value (e.g. control vs. experiment), split the resultant forms in half lengthwise, and position them opposite one another to create a comparative representation of the conditioned distributions (see: https://www.d3-graph-gallery.com/graph/density_mirror.html). In this bivariate application, the comparative symmetry adds value to the analytic process. It is unclear from the paper whether the authors are aware of this namespace convergence, but independent of nomenclature, the paper would benefit from an assessment of the analytic value for making a univariate density plot symmetric.

Of this ambiguous naming of plots, we were not aware. We discuss this ambivalence in the corrected version of the manuscript in the methods section

“It should be noted that there exists an ambiguity in the naming because of the existence of “Mirror Density” plots, a graphical representation of bivariate distributions (e.g., https://www.d3-graph-gallery.com/graph/density_mirror.html). However, maximum likelihood plots can be more informative for such a use case (e.g. (39)). In the opinion of the authors, the name “Mirrored Density” plot (MD plot) is more specific than the “Mirror Density” plot because the density estimation is univariate with a graphical representation as a “line” with a symmetrical reflection of the same information filled out with a color instead of using a bivariate density estimation which does not mirror the “line” of the plot.

- In section 3.5 "The high-dimensional data set (d=45)... is investigated by selecting 12 features": Ridgeline plots with the PDE binning method may be a more space-conservative method of implementing the algorithm (though admittedly, d=45 remains a non-trivial 'curse of dimensionality').

We tried this approach, but for too many features, it becomes infeasible to inspect such a plot. We think that to outline this point is out of the scope of the manuscript.

- With regard to the German stock market data in section 3.5, the schematic MD (Fig. 9) and violin (Fig. 10) plots compare distributions in very different ranges. The paper would benefit from the removal of 'InterestExpense' and 'CapitalExpenditures' from the exemplary features, perhaps to be replaced with features of a range more similar to the other features in the schematic plots.

We thank the reviewer for pointing out this issue. We now are aware of specifying our work more. We added to the results section

“In such a high-dimensional case, statistical testing, parameter settings, usual density plots. and histograms become very troublesome and thus are omitted in this work. Moreover, it becomes challenging to integrate different ranges in one visualization.”

- In general, plots should be ordered where possible, e.g. Fig. 5b,c,d should show skew parameter xi in order [0.6,0.95,1,1.1] for clarity.

The authors thank the reviewer for the suggestion. The mistake was corrected.

- Stacked histograms are not advisable for this application. Stacked histograms make sense when considering how categories sum to a total population (e.g. when exploring various revenue sources and the resultant aggregate revenue in a single graphic). For comparing model distributions of various skewness parameters, e.g. in Fig. 3a, 5a (are these the same histogram?), stacking does not seem appropriate. Neither does stacking seem appropriate for histograms of normalized data, e.g. Robustly Normalized values for Income Tax Share (ITS) and Municipality Income Tax Yield (MTY) in Fig. 12a. The sum total of 2 bins normalized from different ranges provides no substantial comparative analytic value. In both stacked histogram cases: overlaying, rather than stacking, may provide the intended visual effect, and would be appropriate to the data.

The authors thank the reviewer for pointing out this major issue. The manuscript is now rewritten focusing more on ridgeline plots instead of histograms. Histograms are moved to supplementary D because they are challenging to use if many features are given, and the (in)correct binning issue can always be raced. 

The authors thank the reviewer for pointing out the correct definition for stacked histograms. The wording was changed throught the manuscript from “stacked” to “overlayed” because stacked histograms were not computed in the definition of the reviewer above. To be more precise supplementary D has now the sentence:

“Each histogram is computed seperatly and thereafter integrated in one plot using plotly.”

The authors did a mistake by providing both times an overlayed histogram for the skewed experiment instead providing an overlayed histogram for the skewed distribution and one for the bimodal experiment. This is now also corrected.

As a reviewer with experience in data visualization, I feel confident in my assessment of this component of the paper; however, it is my hope that fellow peer reviewers can speak in a more informed manner on the statistical evaluations and experiments performed. The paper's conclusion in Section 5 "current density estimation approaches can lead to major misinterpretations if the default setting is not adjusted" seems to suggest that the scientific community would benefit greatly from the addition of PDE binning methods to existing open-source visualization packages such as ggplot, matplotlib, seaborn and plotly. I hope the authors consider integrated contribution to existing open-source tools.

MD plot was integrated in the ggplot2 syntax and specified in the methods section

“The MD plot can be applied by using installing the R package ‘DataVisualizations’ on CRAN (36) in the framework of ggplot2 (39).”

And in the conclusion section we changed

“The MD plot is available in the R-package ‘DataVisualizations’ on CRAN (33) and in the Python package ‘md_plot’ on PyPi (34). “

To

“The MD plot is available integrated into the R-package ‘DataVisualizations’ on CRAN (36) in the framework of ggplot2, and in the Python package ‘md_plot’ on PyPi (37).

“

Our co-author will try to contact seaborn regarding this idea if our work gets published in a peer-review journal.

The paper would be improved by a spelling check, and a grammar proofread by a native English speaker. Typos and grammatical oversights do not obstruct communication, but do inhibit narrative flow.

The non-native authors are terribly sorry for the inconvenience. We now applied Google’s Grammarly for spell checking. If accepted, Springer Nature could be payed for grammatical corrections and spell-checking.

The Reviewers lead to the following decision by the handling editor Qichun Zhang:

“

PONE-D-19-19081

Analyzing the Fine Structure of Distributions

PLOS ONE

Dear Dr. rer. nat. Thrun,

Thank you for submitting your manuscript to PLOS ONE. After careful consideration, we have decided that your manuscript does not meet our criteria for publication and must therefore be rejected.

I am sorry that we cannot be more positive on this occasion, but hope that you appreciate the reasons for this decision.

Yours sincerely,

Qichun Zhang, PhD

Academic Editor

PLOS ONE

Additional Editor Comments (if provided):

Two reviewers returned the critical comments focusing on the novelty of the manuscript. Basically, the author redo some existing method using Python where the new features have not been demonstrated clearly. In addition, the English writing should be pre-checked where some typos would affect the readability of the manuscript.

“

The authors have shown in this letter that 

 the first reviewer positively reviewed our manuscript, and all suggestions of this reviewer were applied throught the mansuscript

 the second reviewer, as well as apparently the handling editor, did not read the manuscript at all. 

It seems that the handling editor just performed a copy and paste action of the evidently false statements of the second reviewer ignoring the first reviewer's comments except for the last comment regarding the spelling. 

To find out that the statements are false would take less than an hour of work. Any editor should at least be curious about a review consisting of four sentences after a long waiting period of 5 months and check the claims of such a review.

Additionally, minor spelling and grammatical errors should not influence the decision of an editor in a scientific journal at all.

---

## [Decision Letter · Decision Letter 1]

21 Apr 2020

PONE-D-19-19081R1

Analyzing the Fine Structure of Distributions

PLOS ONE

Dear Dr. rer. nat. Thrun,

Thank you for submitting your manuscript to PLOS ONE. After careful consideration, we feel that it has merit but does not fully meet PLOS ONE’s publication criteria as it currently stands. Therefore, we invite you to submit a revised version of the manuscript that addresses the points raised during the review process.

We would appreciate receiving your revised manuscript by Jun 05 2020 11:59PM. To enhance the reproducibility of your results, we recommend that if applicable you deposit your laboratory protocols in protocols.io, where a protocol can be assigned its own identifier (DOI) such that it can be cited independently in the future. For instructions see: http://journals.plos.org/plosone/s/submission-guidelines#loc-laboratory-protocols

We look forward to receiving your revised manuscript.

Kind regards,

Dr Fatemeh Vafaee and Dr David Mayerich

Academic Editors

PLOS ONE

'No'

a) Please provide an amended Funding Statement that declares *all* the funding or sources of support received during this specific study (whether external or internal to your organization) as detailed online in our guide for authors at http://journals.plos.org/plosone/s/submit-now.

b) Please state what role the funders took in the study. If any authors received a salary from any of your funders, please state which authors and which funder. If the funders had no role, please state: "The funders had no role in study design, data collection and analysis, decision to publish, or preparation of the manuscript."

'No'

a. Please update your Competing Interests statement to state any Competing Interests. If you have no competing interests, please state "The authors have declared that no competing interests exist.", as detailed online in our guide for authors at http://journals.plos.org/plosone/s/submit-now

b. This information should be included in your cover letter; we will change the online submission form on your behalf. Please know it is PLOS ONE policy for corresponding authors to declare, on behalf of all authors, all potential competing interests for the purposes of transparency. PLOS defines a competing interest as anything that interferes with, or could reasonably be perceived as interfering with, the full and objective presentation, peer review, editorial decision-making, or publication of research or non-research articles submitted to one of the journals. Competing interests can be financial or non-financial, professional, or personal. Competing interests can arise in relationship to an organization or another person. Please follow this link to our website for more details on competing interests: http://journals.plos.org/plosone/s/competing-interests

4. Please ensure that you refer to Figure 7 in your text as, if accepted, production will need this reference to link the reader to the figure.

Reviewers' comments:

Reviewer's Responses to Questions

**Comments to the Author**

1. If the authors have adequately addressed your comments raised in a previous round of review and you feel that this manuscript is now acceptable for publication, you may indicate that here to bypass the “Comments to the Author” section, enter your conflict of interest statement in the “Confidential to Editor” section, and submit your "Accept" recommendation.

Reviewer #1: (No Response)

Reviewer #4: (No Response)

2. Is the manuscript technically sound, and do the data support the conclusions?

Reviewer #1: Yes

Reviewer #4: Yes

3. Has the statistical analysis been performed appropriately and rigorously? 

Reviewer #1: Yes

Reviewer #4: Yes

4. Have the authors made all data underlying the findings in their manuscript fully available?

Reviewer #1: Yes

Reviewer #4: Yes

5. Is the manuscript presented in an intelligible fashion and written in standard English?

Reviewer #1: Yes

Reviewer #4: Yes

6. Review Comments to the Author

Reviewer #1: The authors have made significant improvements to the manuscript in accordance with reviewer comments. These improvements include the addition of more exemplary data mining applications, new visualization methods with thorough comparative assessments, and careful clarification of novel scientific contribution (which was present in the initial draft, but has been more explicitly declared in the revision).

There are a few minor formatting and language changes needed before publication:

Page 6: “visualizing the b of the estimated probability density distribution (pdf) which will be called in short the distribution of the variable”: If ‘b’ is a variable, it is recommended that it be italicized to avoid confusion.

Grammarly unfortunately doesn’t catch ‘atomic typos’, so be on the lookout for those in final revision. For example, p.11: “The bean plot has a mayor limitation” → “major”, p. 19 acknowledgements: “web scrapping” → ‘web scraping’. Note also p.12: "distrubition” → ‘distribution’.

Please add:

- label to y-axis in Fig. 1a

- y-axis values to Fig. 20, 21

- titles to Fig. 13-18, 25-30

Fig. 10b particularly aids reader understanding of distributional differences, and this reviewer is appreciative of its addition, along with other ridgeline plots and the accompanying assessment of their merits.

The authors’ thoughtful and comprehensive revision of this paper merits its publication.

A note to the editor: It would aid ease of reading for figures and their captions to be included in-context within the paper, with paper body wrapped around. If PLOS intends to increasingly publish content related to data visualization (which would be in the scientific interest), this is a recommended amendment to the paper layout criteria.

Reviewer #4: General comments:

The authors introduce the Mirrored Density plot as a method to automate the visualisation of univariate densities, with a focus on the case where many features from the same dataset need to be visualised. The authors rightly point out that in a situation where the distributions of many variables need to be inspected as part of an exploratory analysis it is crucial that visualisation tools provide robust defaults that avoid producing misleading plots for a wide variety of distributions.

At the core of this manuscript is the authors’ argument that their MD plot, which uses Pareto Density Estimation to obtain a density estimate, is superior to other commonly used visualisations, like the ridgeline, violin, or bean plot. While the argument is generally well presented and the authors offer several examples that are well suited to illuminate the differences between the various visualisation techniques, there is a key point the authors appear to be missing. The process of visualising the distribution of a univariate variable consists of two main steps, density estimation and visualisation. The authors make a convincing argument that PDE is better suited to the task then other commonly used methods, as it doesn’t rely on the user choosing appropriate parameters. However, the authors conflate the issue of density estimation with the visualisation by equating different visualisation approaches with the default estimation techniques offered by the implementations used in the comparison.

The fact that PDE is well suited to the task isn’t particularly surprising but making it readily available for data visualisations is indeed a useful contribution. Considering that the primary contribution of this manuscript is relating to data visualisation (rather than density estimation) I am surprised that they do not offer a more systematic discussion of the relative merits of the different visualisation methods included in the comparison. As I see it the major features that distinguish these plots are

1. Horizontal vs vertical display

2. Presence or absence of a rug

3. Whether density estimates are displayed beyond the range of the data

4. Whether the density estimate is mirrored to create a symmetric display

The authors have chosen a particular combination of these features but do not articulate clearly why they believe this to be desirable nor do they provide any evidence that this particular visualisation (as opposed to density estimation) is superior to others. In fact, it seems to me that the MD plot is essentially a violin plot with different default density estimation.

Detailed comments:

1. The introduction contains several references to histograms that seem less relevant now that histograms have replaced by ridgeline plots for the purpose of the comparison. It would be helpful to shift the focus to ridgeline plots earlier. On pages 3 and 8, it is stated that the comparison will include histograms but there is no mention of the ridgeline plot.

2. I agree that the naming of the plot has potential for confusion with the existing plot of a similar name. The authors may wish to consider whether an alternative name would suit them better. I would, however, discourage arguments about which of the two methods is more appropriately named in the manuscript.

3. The quality of written English in the manuscript is generally acceptable but could be improved in a few places. I would encourage the authors to follow through on their plan to obtain assistance in editing the manuscript prior to publication.

7. PLOS authors have the option to publish the peer review history of their article (what does this mean?). If published, this will include your full peer review and any attached files.

Reviewer #1: Yes: Jane L. Adams

Reviewer #4: Yes: Peter Humburg

---

## [Author Response · Author response to Decision Letter 1]

20 May 2020

Dear Editors Dr Fatemeh Vafaee and Dr David Mayerich,

thank you for having handled our manuscript entitled “Analyzing the Fine Structure of Distributions” and giving us the chance to modify it in order to accommodate the reviewer’s and editor’s comments. 

We have addressed the comments as detailed out in the following (our responses below the reviewers’ comments in red letters). 

PONE-D-19-19081R1

Analyzing the Fine Structure of Distributions

PLOS ONE

Dear Dr. rer. nat. Thrun,

Thank you for submitting your manuscript to PLOS ONE. After careful consideration, we feel that it has merit but does not fully meet PLOS ONE’s publication criteria as it currently stands. Therefore, we invite you to submit a revised version of the manuscript that addresses the points raised during the review process.

We would appreciate receiving your revised manuscript by Jun 05 2020 11:59PM. To enhance the reproducibility of your results, we recommend that if applicable you deposit your laboratory protocols in protocols.io, where a protocol can be assigned its own identifier (DOI) such that it can be cited independently in the future. For instructions see: http://journals.plos.org/plosone/s/submission-guidelines#loc-laboratory-protocols

• A rebuttal letter that responds to each point raised by the academic editor and reviewer(s). This letter should be uploaded as separate file and labeled 'Response to Reviewers'.

• A marked-up copy of your manuscript that highlights changes made to the original version. This file should be uploaded as separate file and labeled 'Revised Manuscript with Track Changes'.

• An unmarked version of your revised paper without tracked changes. This file should be uploaded as separate file and labeled 'Manuscript'.

We look forward to receiving your revised manuscript.

outsourc

Kind regards,

Dr Fatemeh Vafaee and Dr David Mayerich

Academic Editors

PLOS ONE

'No'

a) Please provide an amended Funding Statement that declares *all* the funding or sources of support received during this specific study (whether external or internal to your organization) as detailed online in our guide for authors at http://journals.plos.org/plosone/s/submit-now.

b) Please state what role the funders took in the study. If any authors received a salary from any of your funders, please state which authors and which funder. If the funders had no role, please state: "The funders had no role in study design, data collection and analysis, decision to publish, or preparation of the manuscript."

The authors received no specific funding for this work.

'No'

a. Please update your Competing Interests statement to state any Competing Interests. If you have no competing interests, please state "The authors have declared that no competing interests exist.", as detailed online in our guide for authors at http://journals.plos.org/plosone/s/submit-now

b. This information should be included in your cover letter; we will change the online submission form on your behalf. Please know it is PLOS ONE policy for corresponding authors to declare, on behalf of all authors, all potential competing interests for the purposes of transparency. PLOS defines a competing interest as anything that interferes with, or could reasonably be perceived as interfering with, the full and objective presentation, peer review, editorial decision-making, or publication of research or non-research articles submitted to one of the journals. Competing interests can be financial or non-financial, professional, or personal. Competing interests can arise in relationship to an organization or another person. Please follow this link to our website for more details on competing interests: http://journals.plos.org/plosone/s/competing-interests

The authors state hereby that they have no competing interests.

4. Please ensure that you refer to Figure 7 in your text as, if accepted, production will need this reference to link the reader to the figure.

The manuscript was reformatted accordingly to the suggestions in 1, citations use now the PLOS endnote style and the referencing of the figures is corrected now.

Reviewers' comments:

Reviewer's Responses to Questions

Comments to the Author

1. If the authors have adequately addressed your comments raised in a previous round of review and you feel that this manuscript is now acceptable for publication, you may indicate that here to bypass the “Comments to the Author” section, enter your conflict of interest statement in the “Confidential to Editor” section, and submit your "Accept" recommendation.

Reviewer #1: (No Response)

Reviewer #4: (No Response)

2. Is the manuscript technically sound, and do the data support the conclusions?

Reviewer #1: Yes

Reviewer #4: Yes

3. Has the statistical analysis been performed appropriately and rigorously? 

Reviewer #1: Yes

Reviewer #4: Yes

4. Have the authors made all data underlying the findings in their manuscript fully available?

Reviewer #1: Yes

Reviewer #4: Yes

5. Is the manuscript presented in an intelligible fashion and written in standard English?

Reviewer #1: Yes

Reviewer #4: Yes

6. Review Comments to the Author

Reviewer #1: The authors have made significant improvements to the manuscript in accordance with reviewer comments. These improvements include the addition of more exemplary data mining applications, new visualization methods with thorough comparative assessments, and careful clarification of novel scientific contribution (which was present in the initial draft, but has been more explicitly declared in the revision).

There are a few minor formatting and language changes needed before publication:

1)

Page 6: “visualizing the b of the estimated probability density distribution (pdf) which will be called in short the distribution of the variable”: If ‘b’ is a variable, it is recommended that it be italicized to avoid confusion.

Thanks, this was corrected to:

“This work concentrates on visualizing the estimated probability density distribution (PDF) which will be called the distribution of the variable”

“

Grammarly unfortunately doesn’t catch ‘atomic typos’, so be on the lookout for those in final revision. For example, p.11: “The bean plot has a mayor limitation” → “major”, p. 19 acknowledgements: “web scrapping” → ‘web scraping’. Note also p.12: "distrubition” → ‘distribution’.

Thank you for the valid hint. The noted errors were corrected and professional editing was acquired. It is marked in using the review modus in Word.

2)

Please add:

- label to y-axis in Fig. 1a

- y-axis values to Fig. 20, 21

- titles to Fig. 13-18, 25-30

This is corrected in the revised manuscript.

3)

Fig. 10b particularly aids reader understanding of distributional differences, and this reviewer is appreciative of its addition, along with other ridgeline plots and the accompanying assessment of their merits.

The authors’ thoughtful and comprehensive revision of this paper merits its publication.

A note to the editor: It would aid ease of reading for figures and their captions to be included in-context within the paper, with paper body wrapped around. If PLOS intends to increasingly publish content related to data visualization (which would be in the scientific interest), this is a recommended amendment to the paper layout criteria.

The authors are very grateful to the reviewer for the tedious work making this manuscript way better than it was before.

4)

Reviewer #4: General comments:

The authors introduce the Mirrored Density plot as a method to automate the visualisation of univariate densities, with a focus on the case where many features from the same dataset need to be visualised. The authors rightly point out that in a situation where the distributions of many variables need to be inspected as part of an exploratory analysis it is crucial that visualisation tools provide robust defaults that avoid producing misleading plots for a wide variety of distributions.

At the core of this manuscript is the authors’ argument that their MD plot, which uses Pareto Density Estimation to obtain a density estimate, is superior to other commonly used visualisations, like the ridgeline, violin, or bean plot. While the argument is generally well presented and the authors offer several examples that are well suited to illuminate the differences between the various visualisation techniques, there is a key point the authors appear to be missing. The process of visualising the distribution of a univariate variable consists of two main steps, density estimation and visualisation. The authors make a convincing argument that PDE is better suited to the task then other commonly used methods, as it doesn’t rely on the user choosing appropriate parameters. However, the authors conflate the issue of density estimation with the visualisation by equating different visualisation approaches with the default estimation techniques offered by the implementations used in the comparison.

The authors are grateful for this valid remark demonstrating that different views on this matter exist. To the discussion the following paragraph was added:

“Usually density estimation and visualization approaches are investigated separately from each other. Instead, the authors conflate the issue of density estimation with visualization following the perspective of Tufte, Wilk and Tukey that a graphical representation itself can be used as an instrument for reasoning about quantitative information (8, 53) (p.53). “

To the introduction the following sentence was added:

“On the other hand, “wisely used, graphical representations can be extremely effective in making large amounts of certain kinds of numerical information rapidly available to people” [8], p. 375.”

Please not that in the sentence before, “Moreover” was changed to “On the one hand”. To the conclusion was one sentence added:

“On the other hand, the effective laying open of the data to display the unanticipated, is a major portion of data analysis [8], p. 371.”

Please not that in the sentence before “On the one hand” was added. The authors hope, that these changes indicate that different views on this matter are acceptable.

5)

The fact that PDE is well suited to the task isn’t particularly surprising but making it readily available for data visualisations is indeed a useful contribution. Considering that the primary contribution of this manuscript is relating to data visualisation (rather than density estimation) I am surprised that they do not offer a more systematic discussion of the relative merits of the different visualisation methods included in the comparison.

As I see it the major features that distinguish these plots are

1. Horizontal vs vertical display

The authors wish to thank the reviewer for the chance to improve the manuscript significantly. The discussion states now:

Literature suggests that schematic plots should be wider than they are tall because such shapes usually make it easier for the eye form left to right [2] (p. 129). Small multiples of the type of schematic plots usually present several features with the same graphical design structure at once. Tufte suggests that “If the nature of the data suggests the shape of the graphic follow the suggestion” [52]. Therefore, in the opinion of the authors, the vertical display of box plots [2] should be preferred to the horizontal counterpart of range parts [6], and other schematic plots such as violin plots [3] should be displayed vertically.

6)

2. Presence or absence of a rug

The discussion states now:

Kampstra proposed adding a rug (1D scatter plot) to the violin plot in the bean plot [4]. On the one hand, plotting points in a marginal distribution can easily be misleading [53] (Fig. 1), and the general recommendation is that “the number of information-carrying dimensions (variable) depicted should not exceed the number of dimensions in data” [52] (p.71). On the other hand, if only a handful of unique values are present in the data, then density estimation is inappropriate. Thus, the MD plot does not overlay the density estimation with the 1D scatter plot. Instead, it switches automatically to 1D jittered scatter plots if density estimation results in one or more Dirac delta distributions (e.g., SI F, Fig. 31). The scatter plots are jittered, allowing for a minor indication of the amount of data having one unique value.

7)

3. Whether density estimates are displayed beyond the range of the data

The discussion states now:

The general recommendation is “the larger the share of graphics ink devoted to data, the better, if other relevant matters being equal” [52], (p 96). Tukey and Wilk suggest to avoid undue complexity of form in summarizing and displaying [8], p. 377. Tuftte strongly argues to “erase non-data-ink within reason” [52] (p.96). Hence, the tails of violin-like schematic plots should never extend past the range of data.

8)

4. Whether the density estimate is mirrored to create a symmetric display

Again the authors are very grateful for this valid point. The discussion now states:

“One of the key factors of graphical integrity is to show data variation and not design variation [52]. The schematic plots investigated here have the goal of visualizing such variation by density estimation. Nonsymmetric displays are more useful in the specific task of comparing pairs of distributions to each other. Although bilateral symmetry doubles the space consumed in a graphic without adding new information, redundancy can give context and order to complexity, facilitating comparisons over various parts of data [52] (p.98). The MD plot has the goal of making it easy to compare PDFs, which are often complex. It follows that by using a symmetrical display, clipping, skewness and multimodalities are better visible in data in contrast to nonsymmetrical displays if the body of the symmetric line defined by density estimation is filled out”.

9)

The authors have chosen a particular combination of these features but do not articulate clearly why they believe this to be desirable nor do they provide any evidence that this particular visualisation (as opposed to density estimation) is superior to others.

In fact, it seems to me that the MD plot is essentially a violin plot with different default density estimation.

The authors agree with the reviewer that this issue could be elaborated more in the revised manuscript. Two features of the MDplot were not mentioned in the second draft of the manuscript because the authors thought that a discussion of such thresholds are out of the scope of this manuscript. Now the revised manuscript mentions these features in the methods section:

“The MD plot performs no density estimation below a threshold defining the minimal amount of unique data. Instead, a 1D scatter plot (rug plot) is visualized in which for each unique value, the points are jittered on the horizontal (y-)axis to indicate the number of points per unique value. Another threshold defines the minimal amount of values in the data below which a 1D scatter plot is presented instead of a density estimation. The default setting of both thresholds can be changed or disabled by the user if necessary. These thresholds are advantageous in case of a varying amount missing data per feature or if the benchmarking of algorithms yields quantized error states in specific cases (SI F, Fig. 31).”

To the discussion now the following statements are added:

“In addition to the simple density estimation of several features at once, the MD plot offers several advantages. 1D-scatter plots below a threshold proved very helpful for the benchmarking of clustering algorithms because in several cases, the performance evaluation yielded discrete states (see SI F, Fig 31). To the knowledge of the authors, this has yet to be reported in the literature. The MD plot allows us to investigate distributions after common transformations such as robust normalization and the overlaying of distribution with robustly estimated Gaussians. The usage of transformations is often astonishingly effective [8], p. 376. For example, using the robust transformation in combination with this type of overlaying increased the sensitivity of the tendency that a dataset possesses cluster structures compared to usual statistical testing of the 1st principal component [54]. Wilk and Tukey argued to “plot the results of analysis as a routine matter” [8], p.380, for which the MD plot can be useful tool. For example, ordering features by distribution shapes proved to be helpful if the performance of classifiers is evaluated by cross-validations [55]. If the advantages are combined with the ggplot2 syntax, they provide detailed error probability comparisons [56] with a high data to ink ratio (c.f. [52] (p. 96). 

And a new figure is provided in the supplementary SI F:

“SI F: Exemplary Benchmarking of Cluster Algorithms

Fig. 31.:MD plot for the results the error rate for ten clustering methods are shown on the example of the Lsun3D dataset [58]. It is clearly visible that KM and FKM have two quantized error states contrary to PBC, ProClus and Orclus, for which density has to be estimated. Other methods can only be described by a Dirac delta distribution indicated by a line. The method KM and KM-ID12 differ in the initialization procedure. Abbreviations: KM (k-means), KM-ID12 (specific Initialization procedure), RKM (Reduced k-means), FKM (Factorial k-means), PPC (Projection Pursuit Clustering) with either MD (MinimumDensity), MaximumClusterbility (MC) or NormalisedCut (NC).”

8)

Detailed comments:

1. The introduction contains several references to histograms that seem less relevant now that histograms have replaced by ridgeline plots for the purpose of the comparison. It would be helpful to shift the focus to ridgeline plots earlier. 

Thanks for the remark. Ridge-line plots are now added in the introduction. For completeness. Now also range bars and notched box plots are mentioned shortly:

“ If the goal is to evaluate many features simultaneously, four approaches are of particular interest: the Box-Whisker diagram (box plot) [2], the violin plot [3], the bean plot [4] and the ridgeline plot [5]. The counterparts of the box plot are the range bar [6], and its extension to the notched box plot [7] is nearly unable to visualize multimodality [2]; therefore, it is disregarded in this work“

On pages 3 and 8, it is stated that the comparison will include histograms but there is no mention of the ridgeline plot.

This is corrected now.

9)

2. I agree that the naming of the plot has potential for confusion with the existing plot of a similar name. The authors may wish to consider whether an alternative name would suit them better. I would, however, discourage arguments about which of the two methods is more appropriately named in the manuscript.

The authors follow the suggestion of the reviewer and deleted the following paragraph:

“It should be noted that there exists an ambiguity in the naming because of the existence of “Mirror Density” plots, a graphical representation of bivariate distributions (e.g., https://www.d3-graph-gallery.com/graph/density_mirror.html). However, maximum likelihood plots can be more informative for such a use case (e.g. (43)). In the opinion of the authors, the name “Mirrored Density” plot (MD plot) is more specific than the “Mirror Density” plot because the density estimation is univariate with a graphical representation as a “line” with a symmetrical reflection of the same information filled out with a color instead of using a bivariate density estimation which does not mirror the “line” of the plot.“

However, the reviews process took a long time due to issues for which neither reviewers, current editors, nor authors were responsible. In the passed time, several high-ranking publications were published or are in the process of being published in which the plot with the current naming is used and essential (e.g. see 9). Therefore, the renaming of the plot is unfeasible. 

10)

3. The quality of written English in the manuscript is generally acceptable but could be improved in a few places. I would encourage the authors to follow through on their plan to obtain assistance in editing the manuscript prior to publication.

The authors followed the suggestion of the reviewer and acquired the service of nature springer for language editing. These changes are marked with the review modus in word.

7. PLOS authors have the option to publish the peer review history of their article (what does this mean?). If published, this will include your full peer review and any attached files.

Do you want your identity to be public for this peer review? For information about this choice, including consent withdrawal, please see our Privacy Policy.

Reviewer #1: Yes: Jane L. Adams

Reviewer #4: Yes: Peter Humburg

We hope that the manuscript now meets the criteria for publication in PLOS ONE. We are very much looking forward to hearing from you.

Yours sincerely

Alfred Ultsch, Tino Gehlert and Michel Thrun

---

## [Decision Letter · Decision Letter 2]

14 Jul 2020

PONE-D-19-19081R2

Analyzing the Fine Structure of Distributions

PLOS ONE

Dear Dr. Thrun,

Thank you for submitting your manuscript to PLOS ONE. After careful consideration, we feel that it has merit but does not fully meet PLOS ONE’s publication criteria as it currently stands. We apologize for the delay in getting back to you as we had difficulty in finding enough reviewers for the revised version of your manuscript. Not all the initial reviewers were available to review the revised version and finding a reviewer with relevant expertise who would accept to review has taken a long time.

Nonetheless, we invite you to submit a revised version of the manuscript that addresses the points raised by Reviewer #5. Please submit your revised manuscript by Aug 28 2020 11:59PM. If you will need more time than this to complete your revisions, please reply to this message or contact the journal office at plosone@plos.org. Please include the following items when submitting your revised manuscript:

We look forward to receiving your revised manuscript.

Kind regards,

Fatemeh Vafaee, Ph.D.

Academic Editor

PLOS ONE

Reviewers' comments:

Reviewer's Responses to Questions

**Comments to the Author**

1. If the authors have adequately addressed your comments raised in a previous round of review and you feel that this manuscript is now acceptable for publication, you may indicate that here to bypass the “Comments to the Author” section, enter your conflict of interest statement in the “Confidential to Editor” section, and submit your "Accept" recommendation.

Reviewer #4: All comments have been addressed

Reviewer #5: (No Response)

2. Is the manuscript technically sound, and do the data support the conclusions?

Reviewer #4: Yes

Reviewer #5: Partly

3. Has the statistical analysis been performed appropriately and rigorously? 

Reviewer #4: Yes

Reviewer #5: N/A

4. Have the authors made all data underlying the findings in their manuscript fully available?

Reviewer #4: Yes

Reviewer #5: Yes

5. Is the manuscript presented in an intelligible fashion and written in standard English?

Reviewer #4: Yes

Reviewer #5: Yes

6. Review Comments to the Author

Reviewer #4: (No Response)

Reviewer #5: The authors present a variant of the violin plot, termed “mirrored density” plot, which is intended to provide users with a more useful depiction of the underlying univariate distribution for the purposes of data exploration. The authors correctly highlight the fact that the default parameters for many popular packages may not be suitable for data exploration purposes as they are not sensitive enough to the fine structure of the data. The mirrored density plot is proposed to address this shortcoming of existing visualization software by utilizing Pareto Density Estimation for the estimation of univariate probability densities.

In order to argue for the adoption of mirrored density plots, the authors present a series of experiments on both simulated and real datasets in which mirrored density plots are compared to violin, ridgeline, and bean plots. Statistical tests were performed on simulated datasets to test for the presence of certain assumed/designed features (i.e bimodality and/or skewness), and plots were qualitatively inspected for agreement with these tests.

I commend the authors for reproducing their work in Python in addition to R to ensure that performance is not implementation dependent. Reproducibility is in important concern, and it is good to see the authors taking measures to ensure consistency.

There are several issues that I feel need to be addressed:

1) It is unclear why vioplot was used as the representative package for violin plots. ggplot2 is more widely used and accepted within the R community (155K monthly downloads vs 9K – although ggplot does a lot more than violin plots to be fair).

Furthermore, the underlying functionality for MDplot is provided by ggplot2’s violin plot (https://github.com/Mthrun/DataVisualizations/blob/bc76a8c6dc737cb5c593479a534ef2a5b60b330e/R/ClassMDplot.R#L148), so it seems strange not to use this package for comparison.

Please see Replication_Exp1_Fig3.svg. This figure shows the mirrored density plot overlayed with two violin plots from ggplot2. The green outline was produced with default parameters, and the red line with the minor adjustments which will be described below. When using ggplot2’s violin plot, multimodality is clearly visible when the second mean is 2.4 or 2.5 unlike the plots produced by vioplot.

Please see Replication_Uniform_Fig1.svg. This is a replication of the 1000 uniform samples figure. Again, 2 violin plots are presented on top of the MD plot. The green line is with default parameters. The red line, which almost exactly matches the MD plot, uses kernel=’rectangular’, and adjust=’0.8’. I understand that there is an argument for providing useful default parameters, but I am not convinced it warrants an entirely new package.

The use of vioplot instead of ggplot2 is largely responsible for the author’s claim that “Violin plots in R were not able to visualize the bimodality, which was surprising.”

2) “Statistical testing indicated that the ridgeline plot, bean plot, and MD plot have a similar sensitivity regarding bimodality and skewness as long as the sample is large enough.” – This is a gross misrepresentation of the statistical testing performed in this work. The statistical tests referred to were intended to test for the presence of bi-modality or skewness in the simulated datasets. They do not assess the performance of plotting methods. As such, this should not be taken as statistical evidence supporting MDplot’s performance. This paper is ultimately a qualitative comparison of methods and should be treated as such.

3) There is no justification/discussion regarding sample sizes in the simulated datasets which seem to have been chosen arbitrarily. Why were 1000 samples included for the uniform example, 15500 for multi-modality, and 15000 for skewness?

It would also be valuable to see how each method performs at various sample-sizes as not all data exploration takes place with such a large sample size. The smaller/real dataset experiments do not address this question as the “ground-truth” behind the structure is ultimately unknown.

4) There is no discussion surrounding limitations/shortcomings of the work. It is important to provide this information for potential users so they can make a well-informed decision about whether this package is appropriate for their data. I strongly recommend a discussion surrounding the shortcomings of the qualitative nature of this work. Quantitative comparisons are possible for this sort of work – for example, blind-surveys could be conducted to see whether individuals can detect underlying structure from the plots alone (or whether they detect structure which is not there). Furthermore, there is no discussion surrounding the tendency of this method to over-fit to the data.

Minor corrections:

• Throughout the manuscript, both in-text and in-figures, when referring to a normal distribution, m and sd should be replaced with µ and σ respectively (e.g Fig 3b).

• The authors should attempt to install their package (DataVisualizations) on a clean installation of R. It does not properly install the required packages. These packages have to be added manually.

• It would be nice to have figures either superimposing the MD/violin/bean plots or showing them side by side for an easy visual comparison.

• When referring to the Skewed normal distribution, you should use SN, and not N, to avoid confusion with the actual normal distribution (e.g fig 5b).

• Plots should be formatted consistently. For example, titles of some are bolded (5b) while the others are not (5a).

• Fig 6a uses the naming “beanplot” whereas the rest of the paper uses “bean plot”

• In “Given a feature in the data space, there are several approaches for evaluating univariate structures using the indications of the quantity and range of values, e.g., quantile-quantile plots” – e.g. should just be spelled out as “for example”

• “The counterparts of the box plot are the range bar [7], and its extension to the notched box plot [8] is nearly unable to visualize multimodality [3]; therefore, it is disregarded in this work. ” – I think you mean to say “The boxplot and it’s counterparts (i.e range bar and notched box) are unable to visualize multimodality and are therefore disregarded in this work.”, but I’m not sure.

• The following quote is missing and ending quote: “Pareto density estimation (PDE), the radius for hypersphere density estimation is chosen optimally w.r.t information theoretic ideas [28].

• W.r.t (with respect to) above should probably be spelled out for clarity. Square brackets can be used to indicate a quote has been altered.

• There are several places where the phrase “ridgeline plot, violin plot and bean plot” is used. I would suggest changing to “ridgeline, violin, and bean plots” for brevity. Should this suggestion be ignored, then “Although the Python ridgeline and the violin plot use density estimators implemented in different packages” should be made consistent

• Plot in the quote above should be plots – same correction applies to “in contrast to the histogram and MD plot.”

• Remove the first comma in “The results show that the MD plot is the only schematic plot, which is appropriate for every case and does not require adjustments to its process of density estimation by various parameters” (currently it reads as the MD is the only schematic plot, which is isn’t).

• In “Using web scraping, the information of n=269 cases was extracted.” replace was with were.

Overall, the English is good, however, there are minor typos and punctuation mistakes scattered throughout. I acknowledge that this was professionally vetted by nature/springer for language editing, but they missed several mistakes.

Further notes regarding the similarity of “rectangle” kernel estimation to the Pareto Density Estimation approach:

• I have included another comparison of violin and md plots applied to uniformly sampled data (small_uniform_sample.svg). Again, I was able to get quite a similar result using the “rectangle” option for kernel density estimation.

• In all fairness, this required a smaller bandwidth factor (adjust was set to 0.5 instead of 0.8).

• This suggests that there may be an argument to be made in support of PDE as it does a better job showing the fine-grained structure of the data.

• Furthermore, I concede that these plots taper off towards the end which may be misleading to end users.

---

## [Author Response · Author response to Decision Letter 2]

17 Jul 2020

Responses to reviewers are written in the Rebuttal attached as the cover letter.

---

## [Decision Letter · Decision Letter 3]

4 Aug 2020

PONE-D-19-19081R3

Analyzing the Fine Structure of Distributions

PLOS ONE

Dear Dr. Thrun,

Thank you for submitting your manuscript to PLOS ONE. After careful consideration, we feel that it has merit but does not fully meet PLOS ONE’s publication criteria as it currently stands. Therefore, we invite you to submit a revised version of the manuscript that addresses the points raised during the review process.

We look forward to receiving your revised manuscript.

Kind regards,

Fatemeh Vafaee, Ph.D.

Academic Editor

PLOS ONE

Additional Editor Comments (if provided):

Comment from Editor: I appreciate your effort in improving the manuscript as per reviewers' comments; before accepting the paper, please address minor comment raised by the Reviewer and review the manuscript for English quality making sure that there is no grammatical error and improve figures' quality whenever possible.

Reviewers' comments:

Reviewer's Responses to Questions

**Comments to the Author**

1. If the authors have adequately addressed your comments raised in a previous round of review and you feel that this manuscript is now acceptable for publication, you may indicate that here to bypass the “Comments to the Author” section, enter your conflict of interest statement in the “Confidential to Editor” section, and submit your "Accept" recommendation.

Reviewer #5: All comments have been addressed

2. Is the manuscript technically sound, and do the data support the conclusions?

Reviewer #5: (No Response)

3. Has the statistical analysis been performed appropriately and rigorously? 

Reviewer #5: (No Response)

4. Have the authors made all data underlying the findings in their manuscript fully available?

Reviewer #5: (No Response)

5. Is the manuscript presented in an intelligible fashion and written in standard English?

Reviewer #5: (No Response)

6. Review Comments to the Author

Reviewer #5: I thank the authors for taking the time to consider my recommendations, and I am reasonably satisfied with how they have been addressed. In particular, I am pleased to see a discussion surrounding method limitations and a revision of the interpretation of statistical testing.

Although it would arguably have been more appropriate to compare MD plots to geom_violin (instead of vioplot) in the main figures, the authors have included in-text references to SI F which does make these comparisons and noted the fact that geom_violin is capable of detecting bi-modality.

They have noted that they are happy to improve figures/grammar upon acceptance, so I will leave it to the editor to make a decision regarding this matter.

I will include one small nitpick however. The authors replaced all occurrences of "e.g" with "for example". My apologies for not being more clear with my comments. I was only suggesting that the one instance of e.g be replaced with for example as it was in-sentence. It is still appropriate (and probably preferable) to make use of "e.g" inside parenthesis (e.g here).

---

## [Author Response · Author response to Decision Letter 3]

25 Aug 2020

We asked a second native speaker with appropriate specialization and professional experience to revise the grammar of our manuscript after explaining in detail the content with the goal that corrections are not only grammatically correct but also represent the correct meaning. As we had excellent experience from previous work (https://www.springer.com/gp/book/9783658205393) we hope that the English quality is now considerably improved. All figures have been computed again and then post-processed with Adobe Photoshop. Every change is marked except for supplementary F for which due to technical reasons (Rmarkdown script) the figures could not be post-processed and grammatical corrections are not marked.

The figures now have all the same spelling with regards to the schematic plots, the same font for titles and axis, and as far as possible, the same font size. The various Figures of 1, 3, 5, 6, 8 were combined in two one illustration in which it is noted which subfigure is a), b) and so on. This was achieved via post-processing using Adobe Photoshop. The whole manuscript was revised with regards to grammar.

We are sorry that we did not understand the Reviewer correctly. “e.g.,” is now used inside of parenthesis. The term “for example” was sometimes revised by the native speaker to other options and sometimes remained if there were nor parenthesis. Detail changes are marked in the manuscript.

All figures besides SI F have been uploaded to PACE again and tested there.

---

## [Editor Report · Decision Letter 4]

26 Aug 2020

Analyzing the Fine Structure of Distributions

PONE-D-19-19081R4

Dear Dr. Thrun,

We’re pleased to inform you that your manuscript has been judged scientifically suitable for publication and will be formally accepted for publication once it meets all outstanding technical requirements.

Kind regards,

Fatemeh Vafaee, Ph.D.

Academic Editor

PLOS ONE

---

## [Editor Report · Acceptance letter]

28 Aug 2020

PONE-D-19-19081R4 

Analyzing the Fine Structure of Distributions 

Dear Dr. Thrun:

I'm pleased to inform you that your manuscript has been deemed suitable for publication in PLOS ONE. Congratulations! Your manuscript is now with our production department. 

Kind regards, 

on behalf of

Dr. Fatemeh Vafaee 

Academic Editor

PLOS ONE